# Characteristics and degradation of organic aerosols from cooking sources based on hourly observations of organic molecular markers in urban environments

**Rui Li**[1,2,★], **Kun Zhang**[1,2,★], **Qing Li**[1,2], **Liumei Yang**[1,2], **Shunyao Wang**[1,2], **Zhiqiang Liu**[1,2,3],
**Xiaojuan Zhang**[1,2,3], **Hui Chen**[1,2], **Yanan Yi**[1,2], **Jialiang Feng**[1,2], **Qiongqiong Wang**[4], **Ling Huang**[1,2],
**Wu Wang**[1,2], **Yangjun Wang**[1,2], **Jian Zhen Yu**[5,6], and **Li Li**[1,2]

[1]School of Environmental and Chemical Engineering, Shanghai University, Shanghai, 200444, China
[2]Key Laboratory of Organic Compound Pollution Control Engineering (MOE),
Shanghai University, Shanghai, 200444, China
[3]Jiangsu Changhuan Environment Technology Co., Ltd., Jiangsu, Changzhou, 213004, China
[4]School of Environmental Studies, China University of Geosciences, Wuhan, 430074, China
[5]Department of Chemistry, Hong Kong University of Science and Technology, Hong Kong SAR, 999077, China
[6]Division of Environment and Sustainability, Hong Kong University of Science and Technology,
Hong Kong SAR, 999077, China
★These authors contributed equally to this work.

**Correspondence:** Li Li (lily@shu.edu.cn)

**Abstract.** Molecular markers in organic aerosol (OA) provide specific source information on $PM_{2.5}$, and the contribution of cooking organic aerosols to OA is significant, especially in urban environments. However, the low time resolution of offline measurements limits the effectiveness when interpreting the tracer data, the diurnal variation in cooking emissions and the oxidation process. In this study, we used online thermal desorption aerosol gas chromatography and mass spectrometry (TAG) to measure organic molecular markers in fine particulate matter ($PM_{2.5}$) at an urban site in Changzhou, China. The concentrations of saturated fatty acids (sFAs), unsaturated fatty acids (uFAs) and oxidative decomposition products (ODPs) of unsaturated fatty acids were measured every 2 h to investigate the temporal variations and the oxidative decomposition characteristics of uFAs in urban environments. The average concentration of total fatty acids (TFAs, sum of sFAs and uFAs) was measured to be $105.70 \pm 230.28\,\mathrm{ng\,m^{-3}}$. The average concentration of TFAs in the polluted period ($PM_{2.5} \geq 35\,\mathrm{\mu g\,m^{-3}}$) was $147.06\,\mathrm{ng\,m^{-3}}$, which was 4.2 times higher than that in the clean period ($PM_{2.5} < 35\,\mathrm{\mu g\,m^{-3}}$) and higher than the enhancement of $PM_{2.5}$ (2.2 times) and organic carbon (OC) (2.0 times) concentrations when comparing the polluted period to the clean period. The mean concentration of cooking aerosol in the polluted period ($4.0\,\mathrm{\mu g\,m^{-3}}$) was about 5.3 times higher than that in the clean period ($0.75\,\mathrm{\mu g\,m^{-3}}$), which was similar to the trend of fatty acids. Fatty acids showed a clear diurnal variation. Linoleic acid / stearic acid and oleic acid / stearic acid ratios were significantly higher at dinnertime and closer to the cooking source profile. By performing backward trajectory clustering analysis, under the influence of short-distance air masses from surrounding areas, the concentrations of TFAs and $PM_{2.5}$ were relatively high, while under the influence of air masses from easterly coastal areas, the oxidation degree of uFAs emitted from local culinary sources was higher. The effective rate constants ($k_O$) for the oxidative degradation of oleic acid were estimated to be $0.08$–$0.57\,\mathrm{h^{-1}}$, which were lower than $k_L$ (the estimated effective rate constants of linoleic acid, $0.16$–$0.80\,\mathrm{h^{-1}}$). Both $k_O$ and $k_L$ showed a significant positive correlation with $O_3$, indicating that $O_3$ was the main nighttime oxidant for uFAs in the city of Changzhou. Using fatty acids as tracers, cooking was estimated to contribute an average of 4.6 % to $PM_{2.5}$ concentrations,

increasing to 7.8 % at 20:00 UTC+8 h. Cooking was an important source of OC, contributing 8.1 %, higher than the contribution of $PM_{2.5}$. This study investigates the variation in the concentrations and oxidative degradation of fatty acids and corresponding oxidation products in ambient air, which can be a guide for the refinement of aerosol source apportionment and provide scientific support for the development of cooking source control policies.

## 1  Introduction

Organic aerosol (OA) is an important component of fine particulate matter ($PM_{2.5}$), accounting for 20 %–90 % of the total $PM_{2.5}$ mass (Kanakidou et al., 2005). Among different OA sources, restaurant fumes are relatively important (Huang et al., 2021). The contribution of cooking organic aerosols (COAs) to OA is significant, especially in urban environments, where COAs can contribute 11 %–34 % of total organic carbon (OC) and 3 %–9 % of $PM_{2.5}$ mass concentration, even higher than traffic-related hydrocarbon-based OA (Huang et al., 2021; Li et al., 2020). Carcinogenic mutagens in restaurant fumes contain chemicals that can be harmful to human immune functions (Huang et al., 2020). According to the 2018 global cancer statistics, lung cancer accounts for 24.1 % of all cancer deaths in China and is the most common cause of cancer-related deaths in China. The carcinogenic risk analysis suggested that the potentially adverse health effects induced by cooking sources should not be ignored (N. Zhang et al., 2017).

Cooking is an important source contributor to $PM_{2.5}$, especially in urban environments. Cooking sources have recently received increasing attention, but they are largely an uncontrolled source of $PM_{2.5}$. Saturated fatty acids (sFAs) and unsaturated fatty acids (uFAs), such as palmitic, stearic and oleic acids, are known molecular markers from cooking emissions which are released primarily during cooking activities from the hydrolysis and thermal oxidation of cooking oils. Fatty acids and their derivatives are often used as tracers in the receptor model for the source apportionment of $PM_{2.5}$. It has been found that nonanoic acid, 9-oxononanoic acid and azelaic acid are the main atmospheric oxidation products of oleic acid in the aerosol, while uFAs such as oleic and linoleic acids also react with other atmospheric oxidants, such as hydroxyl (OH) (Nah et al., 2014; Wang et al., 2020).

In previous studies on the molecular tracers of cooking sources based on filter membrane sampling, the time resolution usually varies from 1 d to several days, which cannot accurately capture the diurnal variations in pollutants emitted by the cooking source (Li et al., 2021). Thermal desorption aerosol gas chromatography and mass spectrometry (TAG) enables online monitoring of organic molecular markers (Wang et al., 2020). By clarifying the characteristics of cooking emissions and quantifying the concentrations of pollutants emitted from cooking and their contribution to urban OA on diurnal timescales, we build up data and process knowledge about cooking-source $PM_{2.5}$ pollution, which in turn helps us to evaluate the option of controlling cooking emissions in overall pollution prevention for urban environments.

Processes such as emission rate, atmospheric dilution and photochemical oxidation can affect aerosol composition measured at receptor sites (Fortenberry et al., 2019; Yee et al., 2018). Particulate organic matter can undergo heterogeneous oxidation by ozone, OH and nitrate ($NO_3$) radicals (Wang et al., 2020). When using organic tracer data from filter analysis, variations in concentration due to degradation or secondary production were reported (Ringuet et al., 2012). These degradation and generation processes in the atmosphere are therefore worthy of our attention when using organic markers as source tracers. The mechanism and kinetics of the ozonolysis of oleic acid and linoleic acid in the presence of oxidants such as $NO_3$, $O_3$ and OH radicals have been extensively studied in laboratory studies (Vesna et al., 2009; Zahardis and Petrucci, 2007; Ziemann, 2005). The aging of primary organic aerosol (POA) markers under atmospheric conditions, however, is still far from being properly understood with few field observations performed on this topic compared to laboratory studies (Bertrand et al., 2018a, b). Highly time-resolved observations would help to fill this gap.

In this study, TAG was employed at an urban site in Changzhou, China, to investigate the variation in atmospheric cooking-related fatty acids with hourly-resolution data (Ren et al., 2019). The aim of this study is to identify the contribution of cooking emissions to ambient $PM_{2.5}$ with hourly organic molecular data and to investigate the oxidative decomposition reactions of cooking-related uFAs in an urban area. Results of this study could provide a valid basis and insights for the refinement of $PM_{2.5}$ source apportionment, as well as atmospheric modeling.

## 2  Methodology

### 2.1  Field measurement

Gaseous pollutants, $PM_{2.5}$ and its main chemical constituents (water-soluble ions, carbon components, elements, etc.), as well as organic markers (alkanes, hopanes, polycyclic aromatic hydrocarbons, sugars, alcohols, organic acids, etc.), were measured online at the Changzhou Environmental Monitoring Center of Jiangsu Province (CEMC) (31.76° N,

119.96° E) during January–March 2021, which is a representative urban site (Fig. 1). The meteorological parameters were obtained from a meteorological monitor (WXT520, VAISALA Inc., FL). $O_3$ and $NO_2$ were measured with an ozone analyzer (49i-PS, Thermo Fisher Scientific, US) and a NOx analyzer (MODEL42i, Thermo Fisher Scientific, US), respectively. $PM_{2.5}$ mass concentration was measured with an online particulate matter monitor (BAM1020. Met One Inc., US). The concentration of the carbon components (organic carbon, OC; elemental carbon, EC) was measured using semi-continuous OC–EC analyzer (RT-4, Sunset Laboratory Inc, US) (Nicolosi et al., 2018; Q. Zhang et al., 2017). Water-soluble ions were measured with a MARGA ionic online analyzer (ADI2080, Metrohm, Switzerland) (Makkonen et al., 2012), and elements were measured with an atmospheric elements online monitor (EHM-X200, Tianrui, China).

The quantification of hourly speciated organic markers was achieved using TAG. The operation details and data quality have been described in our previous work (Wang et al., 2020; Zhang et al., 2021). The sampling and analysis sequence of the TAG system includes four steps: (a) $PM_{2.5}$ sampling and synchronous gas chromatography–mass spectrometry (GC-MS) analysis of the previous sample; (b) loading of the internal standards (ISs) from the standard (STD) reservoir to a thermal desorption cell; (c) derivatization and thermal desorption of analytes on the collection and thermal desorption (CTD) cell and subsequent preconcentration of the analytes in a focusing trap (FT); and (d) loading of analytes into the GC column for GC-MS analysis. The following is a detailed description. Ambient air was sampled at a flow rate of 8.5–9.5 L min$^{-1}$ through a cyclone with $PM_{2.5}$ cutting size (BGI Inc., Waltham, MA), a Nafion dryer (PERMA PURE, MD-700-24S-3) to remove moisture and then a carbon denuder (model: ADI-DEN2) to remove volatile organics. The sampled particles were collected on the CTD cell at 30 °C for 60 min, followed by derivatization and thermal desorption for 8 min as the temperature of the CTD cell increases to 300 °C in 2 min and is maintained for 6 min, during which a 10 mL min$^{-1}$ helium purge flow combined with a 40 mL min$^{-1}$ derivatization flow with N-methyl-N-(trimethylsilyl)trifluoroacetamide (MSTFA) flows through for 8 min. Subsequently, the FT was heated to 300°C in 2 min and kept at 300°C for 10 min, transferring the analytes onto the GC column head (DB-5MS, size 30 m × 0.25 μm × 0.25 μm) by carrier gas. After GC separation, the target organics were sent to the MS detector for quantification. The GC-MS analysis duration for each sample was 60 min while the collection of the next sample of the CTD cell starts. With the current TAG instrumental setup, samples were collected every even hour. The post-sampling steps, including in situ derivatization, thermal desorption, GC-MS analysis and standby step, took 2 h, thus producing 12 samples per day.

The summary of target organic molecular markers and ISs is shown in Table 1. The identification of compounds was performed by comparing retention times and mass spectra with those of authentic standards (Vesna et al., 2009; Wang et al., 2020). Calibration curves were established by the internal standard method. The correlation coefficients of the calibration curves range from 0.88–1.00. For compounds without authentic standards and for compounds whose authentic standards are not included in the current standard mixture, their identification is performed by comparing their mass spectra with the National Institute of Standards and Technology (NIST) libraries. Azelaic acid was identified and quantified by using authentic standards. Nonanoic acid and 9-oxononanoic acid were identified by comparison with mass spectra in the NIST library and by referring to Ziemann (2005), Pleik et al. (2016), and Wang et al. (2020). Ozone oxidation of oleic acid yields $C_9$ aldehydes and acids including nonanal, azelaic acid, nonanoic acid and 9-oxononanoic acid. Since nonanal could also be primarily in the gas phase, it is thus not discussed in this paper. The library of the NIST was identified and quantified using the alternative standards specified in Table 1.

## 2.2 Backward trajectory analysis

Backward trajectory analysis is a useful tool in identifying the influence of air masses on the chemical composition of $PM_{2.5}$ (Wang et al., 2017). Backward trajectories of 36 h duration arriving at an altitude of 100 m above ground level (a.g.l.) over the CEMC site were calculated deploying 0.5° Global Data Assimilation System (GDAS) meteorological data (https://www.ready.noaa.gov/archives.php, last access: 16 August 2022). The trajectories were then classified into different clusters according to the geographical origins and movement of the trajectories using the TrajStat model (Li et al., 2020).

## 2.3 Relative rate constant analysis

Ambient concentrations of species are influenced by the emissions, atmospheric dilution/compaction, chemical loss/production and wet/dry deposition. As the target sFAs and uFAs in urban environments are predominately primary in their source origin, the chemical production rate could be assumed to be negligible. Donahue et al. (2005) formulated the relative rate expression for heterogeneous oxidation reactions of multicomponent OA. The specific expression applied to the ambient measurements of uFAs is derived as in Eq. (1) and Eq. (2) (Wang and Yu, 2021).

$$\frac{C_i}{C_s} = A \times e^{-kt} \tag{1}$$

$$k \approx k_{r_i} \times C_{OX} \tag{2}$$

$C_i$ and $C_s$ are the particle-phase concentration of species $i$ and sFAs, respectively. Among the quantified sFA and uFA

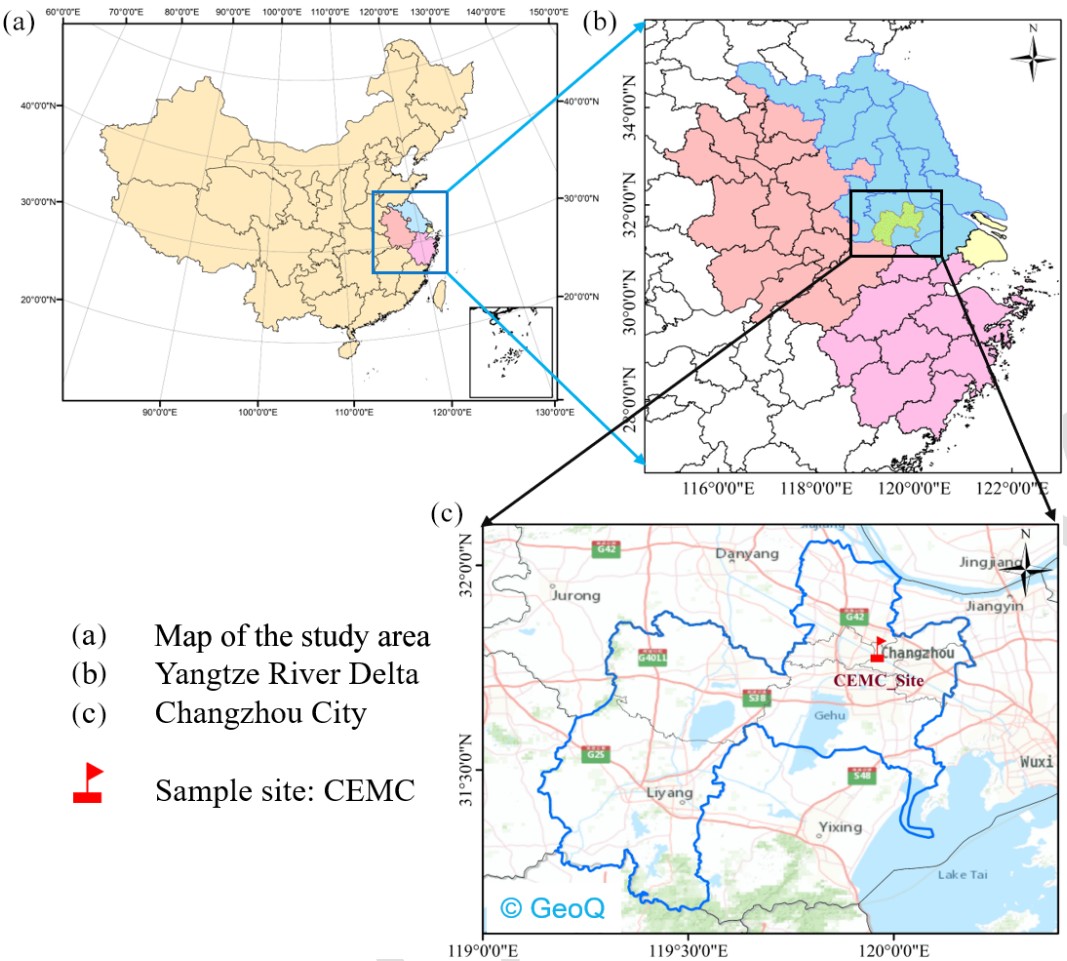

(a)      Map of the study area
(b)      Yangtze River Delta
(c)      Changzhou City

     Sample site: CEMC

**Figure 1.** Location of the sampling site in Changzhou, China.

**Table 1.** Statistics of hourly concentrations of organics associated with cooking emissions measured by TAG during the campaign.

| Compounds | Average | SD | Min | Max | Quantification IS |
|---|---|---|---|---|---|
| Myristic acid[a] | 0.69 | 1.33 | 0.03 | 10.14 | Palmitic acid-$d_{31}$ |
| Palmitic acid | 38.77 | 84.14 | 1.45 | 670.12 | Palmitic acid-$d_{31}$ |
| Stearic acid | 26.51 | 50.58 | 1.81 | 341.65 | Palmitic acid-$d_{31}$ |
| Oleic acid | 32.15 | 81.34 | 0.96 | 723.95 | Stearic acid-$d_{35}$ |
| Linoleic acid[b] | 7.80 | 28.32 | 0.09 | 326.50 | Stearic acid-$d_{35}$ |
| Nonanoic acid[c] | 1.19 | 1.32 | BD[d] | 7.94 | Adipic acid-$d_{10}$ |
| 9-oxononanoic acid[c] | 3.91 | 4.73 | 0.19 | 17.18 | Adipic acid-$d_{10}$ |
| Azelaic acid | TSI 9.15 | 32.99 | BD | 309.64 | Adipic acid-$d_{10}$ |

[a] Quantified using palmitic acid as the surrogate. [b] Quantified using oleic acid as the surrogate.
[c] Quantified using azelaic acid as the surrogate. [d] Below detection limit.

cooking markers, palmitic acid was selected as the reference molecule for normalization. Using the concentration ratio eliminates the interference from atmospheric dilution and deposition. Fitting the ambient $C_i/C_s$ data versus $t$ with an exponential function provides an estimate for $k$, the effective pseudo-first-order decay rate (h$^{-1}$). $k_{ri}$ is the second-order reaction rate constant of species $i$ against an oxidant. $C_{OX}$ is the average oxidant concentration in the aerosol.

## 2.4   Source apportionment based on PMF

Positive matrix factorization (PMF) is a bilinear factor analysis method which is widely used to identify pollution sources

and quantify their contributions to the ambient air pollutants at receptor sites, with an assumption of mass conservation between emission sources and receptors (Amato et al., 2009; Lee et al., 2008). In this study, the United States Environmental Protection Agency (USEPA) PMF version 5.0 (Norris et al., 2014) was applied to perform the analysis. PMF decomposes the measured data matrix, $X_{ij}$, into a factor profile matrix, $f_{kj}$, and a factor contribution matrix, $g_{ik}$ (Eq. 3):

$$x_{ij} = \sum_{k=1}^{p} g_{ik} f_{kj} + e_{ij}, \tag{3}$$
$$Q = \sum_{i=1}^{n} \sum_{j=1}^{m} (e_{ij}/u_{ij})^2, \tag{4}$$

where $X_{ij}$ is the measured ambient concentration of target pollutants, $g_{ik}$ is the source contribution of the $k$th factor to the $i$th sample, $f_{kj}$ is the factor profile of the $j$th species in the $k$th factor, and $e_{ij}$ is the residual concentration for each data point. PMF seeks a solution that minimizes an object function $Q$ (Eq. 4), with the uncertainties of each observation ($u_{ij}$) provided by the user.

The uncertainty of each data point was calculated according to Eq. (5):

$$u_{ij} = \sqrt{\left(x_{ij} \times \text{EF}\right)^2 + \left(\frac{1}{2} \times \text{MDL}\right)^2}, \tag{5}$$

where MDL is the method detection limit and EF is the error fraction determined by the user and associated with the measurement uncertainties. The concentration data below MDL were replaced by half of the MDL, and the corresponding uncertainty $u_{ij}$ was calculated as five-sixths of the MDL. Missing values were replaced by the median value of the species, and its $u_{ij}$ was assigned as 4 times the median value (Norris et al., 2014).

## 3 Results and discussion

The time series of hourly data of meteorological parameters, gaseous pollutants (including $O_3$ and $NO_2$), $PM_{2.5}$, water-soluble ions and carbon components during the monitoring period (10–14 January, 9–15 February and 11–16 March 2021) are shown in Fig. 2. During the campaign, the average temperature ($T$), relative humidity (RH) and wind speed (WS) were $10.9 \pm 4.5°$, $55.3 \pm 18.2\%$ and $1.2 \pm 0.5 \, \text{m s}^{-1}$, respectively. The average concentrations of gas pollutants, $PM_{2.5}$, water-soluble ions, and OC and EC are listed in Table S2. The average concentrations of $NO_2$, $O_3$ and $PM_{2.5}$ were $42.85 \pm 25.89$, $51.53 \pm 29.62$ and $50.07 \pm 26.54 \, \mu\text{g m}^{-3}$, respectively. Additionally, the average OC and EC concentrations were $6.57 \pm 4.63$ and $2.12 \pm 2.04 \, \mu\text{g m}^{-3}$, respectively, with the contribution of OC to $PM_{2.5}$ ranging from 4.7 % to 26.8 % (13.2 % as average).

## 3.1 Characteristics of cooking-derived organic molecular markers

The fatty acids studied include the three most abundant sFAs (myristic, palmitic and stearic acids) and the two most abundant uFAs (oleic and linoleic acids). The concentration of total fatty acids (TFAs; sum of the concentrations of the five fatty acids) was $105.70 \pm 230.28 \, \text{ng m}^{-3}$, ranging from 8.30 to $2066.30 \, \text{ng m}^{-3}$, which is close to the concentrations at the urban site in Shanghai ($105 \, \text{ng m}^{-3}$) (Li et al., 2020; Wang et al., 2020). The average percentage of TFAs in OC was 1.3 % with the maximum value of 8.7 % (the concentration of $PM_{2.5}$ at the corresponding time was $99 \, \mu\text{g m}^{-3}$), which was 6.6 times higher than the average. It revealed that the composition of $PM_{2.5}$ could dramatically change, especially during dinnertime (18:00–20:00; all times are UTC+8 h). The mean concentration of TFAs at dinnertime was $172.89 \, \text{ng m}^{-3}$, and the contribution of TFAs to $PM_{2.5}$ and OC mass concentration was 2.7‰ and 1.8 %, respectively, which were 1.6 and 1.4 times the mean during the observation period.

We define the "polluted period" as the periods with hourly $PM_{2.5}$ concentrations exceeding $35 \, \mu\text{g m}^{-3}$, and the remaining periods are defined as the "clean period". Table 2 shows the mean values of $PM_{2.5}$, OC, TFA concentrations and meteorological conditions during the clean ($PM_{2.5} < 35 \, \mu\text{g m}^{-3}$) and polluted periods ($PM_{2.5} \geq 35 \, \mu\text{g m}^{-3}$). Generally, the meteorological conditions during the polluted period are unfavorable compared to the clean period, showing lower wind speed and higher humidity. The ratios of WS, $T$ and RH during the polluted period to the clean period are 0.9, 1.0 and 1.1, respectively. The mean concentration of $PM_{2.5}$ during the polluted period was $62.86 \, \mu\text{g m}^{-3}$, which was 2.2 times higher than that during the clean period ($28.29 \, \mu\text{g m}^{-3}$). OC and $PM_{2.5}$ were similar, with concentrations during the polluted period being 2.0 times higher than during the clean period. The mean concentration of TFAs in the polluted period was $147.06 \, \text{ng m}^{-3}$, 4.2 times higher than that in the clean hours ($35.28 \, \text{ng m}^{-3}$). Additionally, the concentrations of sFAs and uFAs in the polluted hours were 4.3 and 4.1 times higher than those during the clean period, respectively.

The concentrations of TFAs were influenced by emissions, accumulation, transport and dispersion of pollutants during the polluted periods (Hou et al., 2006; Schauer et al., 2003). The fatty acid content of $1.95 \, \text{ng} \, \mu\text{g}^{-1}$ in $PM_{2.5}$ during the polluted period was 1.6 times greater than that of $1.24 \, \text{ng} \, \mu\text{g}^{-1}$ during the clean period, which was smaller than the variation range of $PM_{2.5}$ and OC concentrations before and after the polluted period. The variation in TFAs in OC was similar to that in $PM_{2.5}$. Table S1 in the Supplement shows the contribution of total fatty acids directly emitted from various sources to OC, in which the contribution of TFAs from vehicle exhaust is the least and the proportion of TFAs emitted from cooking in OC is higher than that from other sources. The change in TFAs / OC was weaker than the change in OC mainly because cooking has relatively small

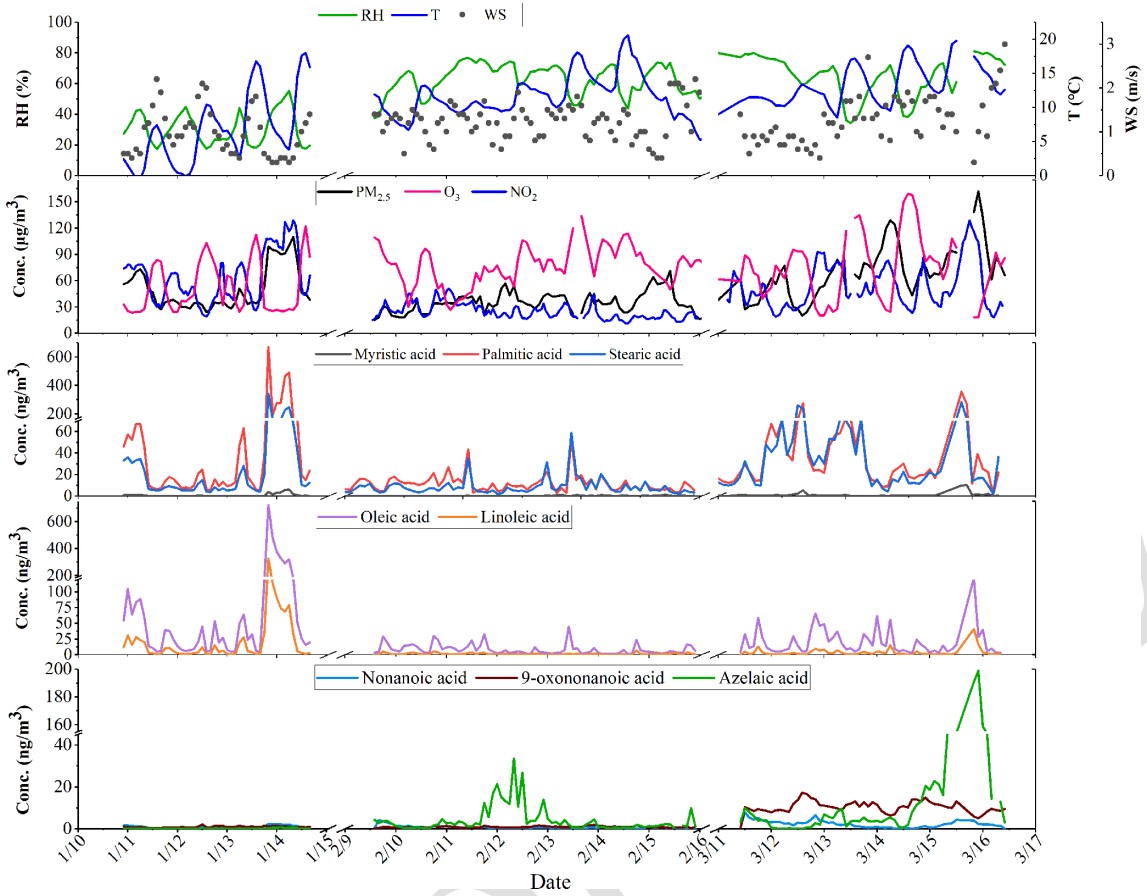

**Figure 2.** Time series of pollutant concentrations and meteorological parameters.

fluctuations in emissions, while the increase in OC concentration was more significant with simultaneous contributions from other sources (e.g., biomass burning, coal combustion and vehicle exhaust). Similarly, the mass concentration of $PM_{2.5}$ was significantly driven by emission sources. The observed contribution of TFAs to OC in $PM_{2.5}$ was smaller than the TFAs / OC ratio in cooking but larger than that in other sources.

Similar variation and diurnal patterns were found for these five fatty acids (Fig. 3), confirming their common origin. In addition, compared to fatty acids, the time series of $C_9$ acids showed a different diurnal variation, suggesting different production and reaction processes. Fatty acids showed a clear diurnal variation, with two peaks observed at around 06:00 and 20:00 local time, respectively, and the dinnertime peak was especially prominent. In contrast to the previous observations in Shanghai, no peak was observed at lunchtime. The relatively higher boundary layer during the daytime facilitated the diffusion of pollutants. The weaker oxidation of uFAs emitted at night made the fatty acid concentration peaks more pronounced at dinnertime (Wang et al., 2020). Figure 3b shows the contribution of various fatty acids to OC. From the diurnal patterns, it is shown that the

**Table 2.** $PM_{2.5}$ concentration, organic carbon fraction, fatty acid concentration and meteorological conditions during the clean and polluted periods.

| Species | Clean period | Polluted period | Polluted/ clean |
|---|---|---|---|
| $PM_{2.5}$ ($\mu g\,m^{-3}$) | $28.29 \pm 5.27$ | $62.86 \pm 25.67$ | 2.2 |
| OC ($\mu g\,m^{-3}$) | $4.05 \pm 1.09$ | $8.00 \pm 5.23$ | 2.0 |
| TFAs ($ng\,m^{-3}$) | $35.28 \pm 28.17$ | $147.06 \pm 281.66$ | 4.2 |
| sFAs ($ng\,m^{-3}$) | $21.60 \pm 14.91$ | $92.05 \pm 162.75$ | 4.3 |
| uFAs ($ng\,m^{-3}$) | $13.68 \pm 14.22$ | $55.53 \pm 133.82$ | 4.1 |
| TFAs / $PM_{2.5}$ ($ng\,\mu g^{-1}$) | $1.24 \pm 0.91$ | $1.95 \pm 2.85$ | 1.6 |
| TFAs / OC ($ng\,\mu g^{-1}$) | $9.52 \pm 7.79$ | $15.33 \pm 14.59$ | 1.6 |
| WS ($m\,s^{-1}$) | $1.23 \pm 0.45$ | $1.14 \pm 0.54$ | 0.9 |
| $T$ (°) | $10.77 \pm 4.22$ | $10.99 \pm 4.68$ | 1.0 |
| RH (%) | $53.41 \pm 17.49$ | $56.33 \pm 18.55$ | 1.1 |

proportion of the five fatty acids and TFAs in OC at noon had a weaker peak, which was still smaller than that during the morning and evening mealtimes. In conclusion, the apparent peaks of TFAs at dinnertime provide evidence for major source contribution to air pollution from local cooking emissions, although there are a mix of sources including vehicle exhaust, coal combustion, etc.

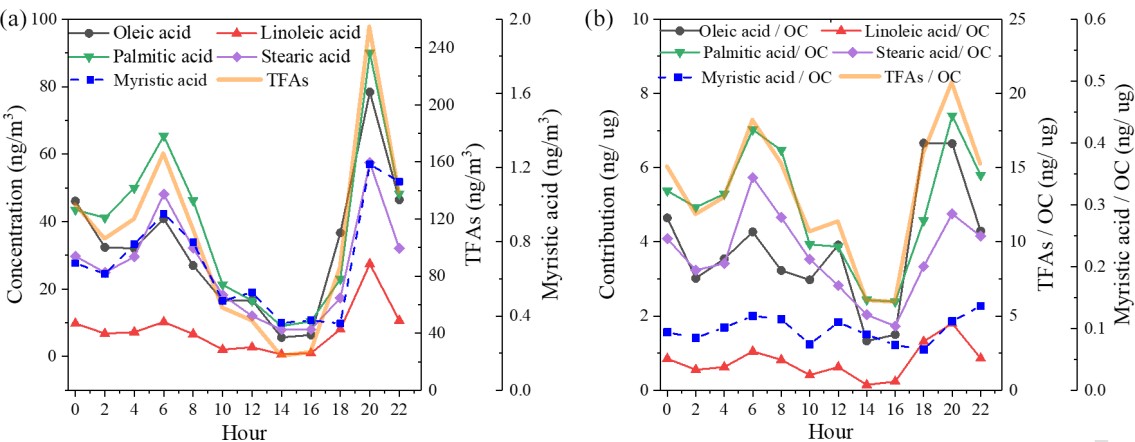

**Figure 3.** Diurnal variation in five fatty acids and TFAs during the observation period.

Fatty acids in urban atmospheres are influenced by various anthropogenic (e.g., biomass burning, vehicle exhaust) (Hays et al., 2002; Schauer et al., 2001; Simoneit, 2002; Wang et al., 2009) and biogenic sources (Oliveira et al., 2007; Rogge et al., 2006). The main sources of fatty-acid-like substances in the ambient air of the study area can be discerned on the basis of characteristic ratios between fatty acids emitted from different sources (Fig. 4) (He et al., 2004; Pei et al., 2016; Rogge et al., 1993; Zhao et al., 2015, 2007). The palmitic acid to stearic acid (P/S) ratios observed in this study range between 0.49 and 3.08 (average value: 1.49), significantly lower than those associated with residential coal combustion and industrial coal combustion while partially overlapping those from biomass burning, vehicle exhaust and sea spray aerosol (Bikkin et al., 2019; Cai et al., 2017; Ho et al., 2015; Zhang et al., 2008, 2007). Ho et al. (2015) investigated urban areas in Beijing where fatty acid concentrations were elevated during traffic restrictions compared to non-restricted periods, suggesting that motor vehicle exhaust is not the largest source of fatty acids in urban areas. The information on fatty acid emissions from biomass burning sources is closer to that of cooking sources (Hays et al., 2002; Schauer et al., 2001; Zhang et al., 2008); however, in the study of Simoneit (2002), no oleic acid was detected in organic molecular substances from biomass burning. The oleic acid / stearic acid (O/S) ratio from sea spray aerosol samples is 0.16 (Bikkin et al., 2019), which is obviously lower than the ambient data in this study (1.4). Hence, during the observation in this study, vehicle exhaust and sea spray were not the most important sources of fatty acid emissions in urban Changzhou, especially during the dinner period, when the O/S ratio was significantly higher and close to the ratio of the organics emitted from traditional culinary types in the Yangtze River Delta region.

Information on the changes in specific molecular markers is useful in investigating the aging process of aerosol. The two uFAs (oleic acid and linoleic acid) are more reactive with atmospheric oxidants (OH and $O_3$, etc.) in the atmosphere due to the presence of C=C bonds compared to sFAs. Furthermore, the two homologous sFAs (palmitic and stearic acid) have similar chemical structures, reactivity and volatility; thus their concentration ratios can be assumed to remain constant during post-emission periods. Therefore, the ratio of P/S mainly depends on the sources. Figure 5 shows the O/S ratios and linoleic acid / stearic acid (L/S) versus P/S, respectively. The average value of P/S was $1.49 \pm 0.49$, which was within the range of cooking source profile values measured from direct emissions from different restaurants and cooking types (1.3–8.1) (He et al., 2004; Pei et al., 2016; Schauer et al., 2002; Zhao et al., 2007), as well as similar to the ratio of P/S in atmospheric $PM_{2.5}$ in Shanghai (1.9) (Li et al., 2020; Wang et al., 2020). In this study, the O/S ratio ($1.4 \pm 1.1$) of the ambient samples was overall in the range of the cooking source profile ($3.6 \pm 1.6$), while the L/S ratio of $0.25 \pm 0.31$ was slightly lower than the cooking source profile values ($2.9 \pm 1.8$) (He et al., 2004; Pei et al., 2016; Schauer et al., 2002; Zhao et al., 2007). The results of Moise and Rudich (2002) showed that the reactant activity is directly related to the concentration of unsaturated bonds, with linoleic acid having an extra double bond compared to oleic acid, indicating that linoleic acid is more easily degraded than oleic acid (Moise and Rudich, 2002; Thornberry and Abbatt, 2004). The O/S ratio of the ambient samples in this study was higher than those measured in Beijing (0.65) (He et al., 2004) from January to October and in Shanghai (0.83) (Li et al., 2020; Wang et al., 2020) during winter.

The diurnal variations in O/S and L/S are also shown in Fig. 5. The ratios were significantly higher during dinnertime and were closer to the cooking source profile, demonstrating that changes in fatty acid concentrations may be influenced primarily by cooking source emissions, especially during dinnertime when fresh cooking source emissions entered into the atmosphere and uFAs were quickly consumed due to aging. The ratio of linoleic acid to stearic acid is consistently

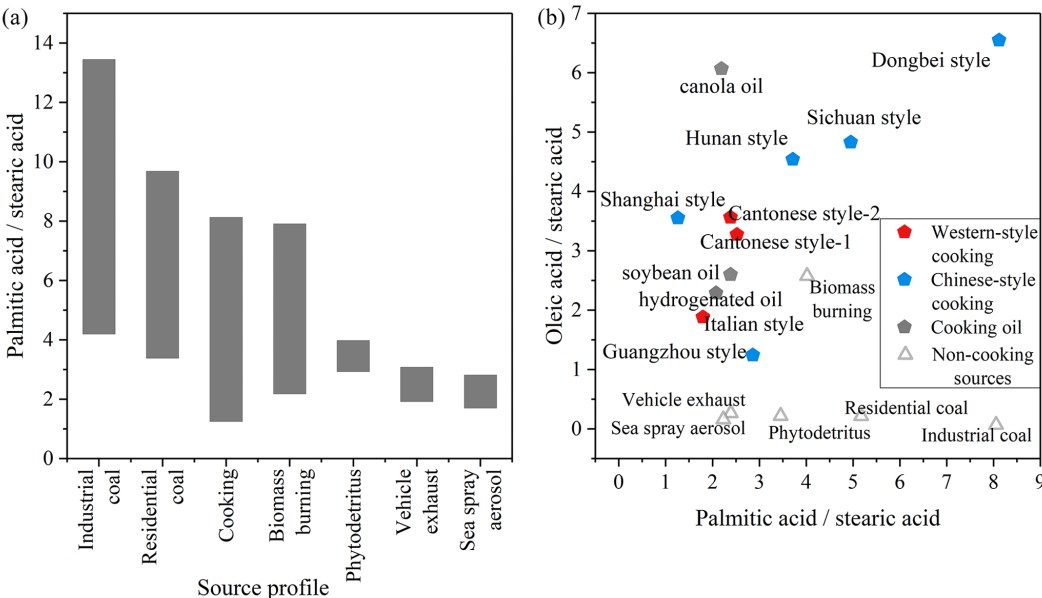

**Figure 4.** Ratio of fatty acids (P/S) in organic molecular substances emitted directly from different sources **(a)** and ratio of fatty acids (P/S versus O/S) emitted by different types of cooking sources and non-cooking sources **(b)** (Bikkin et al., 2019; Cai et al., 2017; Hays et al., 2002; He et al., 2004; Oliveira et al., 2007; Pei et al., 2016; Rogge et al., 1993; Schauer et al., 2001, 2002; Simoneit, 2002; Zhang et al., 2008; Zhao et al., 2007).

lower than what is involved in the source spectrum, which may be influenced by different regions and source characteristics from different types of restaurants, as well as small emissions from other minor sources.

## 3.2 Backward trajectory clustering analysis

The best solution of four clusters was determined based on the variation in the total spatial variance (Figs. 6 and S2). Figure 7 shows the four cluster solutions and the mean distribution of meteorological conditions and pollutants in each cluster. Briefly, cluster 1 (CL#1), which represents 15.4 % of the sample, comes from the northwest continental region of China and reaches Changzhou before passing Gansu, Shan'xi and Henan provinces, and the lower temperature and humidity associated with this cluster are consistent with its geographic origin. Cluster 2 (CL#2), which accounts for 35.6 % of the total number of trajectories, represents air masses from the northeastern part of the ocean, and the temperature and humidity associated with this cluster are higher than those of CL#1. Cluster 3 (CL#3), contributing 18.6 % and traveling slowly from inland area, is associated with the lowest wind speed, with higher temperature and humidity than CL#1 but lower than CL#2. Cluster 4 (CL#4), representing 30.3 % of the trajectories, represents the eastern and southeastern oceanic air masses, with the highest observed temperature, humidity and wind speed among all of the air masses. CL#2 and CL#4 have relatively high temperature, humidity and wind speed. CL#3 is associated with the high-

est NO$_2$ concentrations, confirming its local air mass origin, and the PM$_{2.5}$ and OC concentrations in this air mass are also the highest compared to all the other air masses.

The concentrations of sFAs and uFAs and their oxidation products in each cluster are shown in Fig. 6. The total concentrations of the oxidative decomposition products of sFAs, uFAs and ODPs (oxidative decomposition products; in this study, ODPs include azelaic acid, nonanoic acid and 9-oxononanoic acid) within the four types of air mass clusters were in the following order: CL#3>CL#2>CL#4>CL#1, where the TFAs in CL#1 and CL#3 were larger than the percentages in CL#2 and CL#4. The relative contents of sFAs and uFAs in CL#1 and CL#3 are closer than those in the other two types of air masses and are closer to the concentration ratio of the species directly emitted from the cooking source (the values of uFAs / sFAs range from 0.8 to 3.2) (He et al., 2004; Pei et al., 2016; Schauer et al., 2002; Zhao et al., 2007), which indicated that the oxidative decomposition of uFAs is less in CL#1 and CL#3. CL#3 was a slowly moving, local cluster. Under this air mass clustering, local emissions contribute significantly to fatty acids, as well as PM$_{2.5}$ concentration. The air mass of CL#1 exhibits the longest range, the concentrations of ODPs were relatively small among all air masses, and the low ODP concentration was inconsistent with other literature findings of more aging aerosol production from long-range transport (Wang et al., 2020). The lowest PM$_{2.5}$ concentration and cleaner air masses during air mass CL#1 suggested that long-range air mass transport from the northwest was not the main source of fatty acids

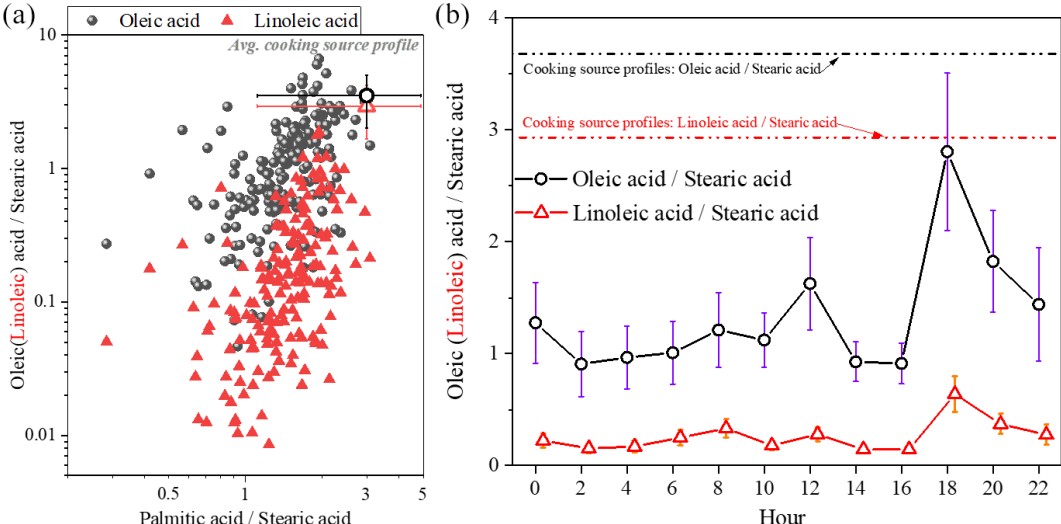

**Figure 5.** The oleic / stearic acid and linoleic / stearic acid ratios compared to the palmitic / stearic acid ratio **(a)** and diurnal variation in the ratio of oleic (linoleic) acid to stearic acid concentration **(b)** (the cooking source profile values were measured from direct emissions from different restaurants and cooking types) (He et al., 2004; Pei et al., 2016; Schauer et al., 2002; Zhao et al., 2007).

and ODPs in Changzhou during the observation. The value of the uFAs / sFAs ratio in CL#2 and CL#4 was less than that in CL#1 and CL#3 and less than the ratio in the emission sources. In addition, the proportion of ODPs in CL#2 and CL#4 is greater than that in CL#1 and CL#3. This result may be explained by the following two reasons: first, under the influence of transport, the air masses brought more sFAs and ODPs, and the air masses were more aged; for example, marine heterotrophic bacteria releases sFAs and uFAs into the water column. However, the mono- and polyunsaturated fatty acids (e.g., oleic acid, linoleic acid) in seawater rapidly oxidize to form initially oxocarboxylic acids, azelaic acid, etc. (Bikkin et al., 2019). Second, under the influence of CL#2 and CL#4 air masses, in which the ozone concentration was higher than other air masses, the decomposition reaction of uFAs was more active and could produce more ODPs. In addition, the oxidative reaction of uFAs could be influenced by meteorological conditions as well.

## 3.3 Atmospheric aging of unsaturated fatty acids

Figure 8a shows the diurnal variation in ozone, oleic acid and ODPs. The ozone concentration started to rise in the morning (06:00) and peaked in the late afternoon (14:00). The diurnal trend of oleic acid was opposite to that of ozone. The diurnal trend of ODPs was also different from oleic acid, and the small peak of ODPs was found at around 12:00 in the daytime, which was earlier than that of ozone. At the same time, oxidative decomposition, atmospheric dilution and lower emissions caused a significant decrease in the concentration of oleic acid until night when large amounts of fresh emissions enter the atmosphere again. The decreasing rate of oleic acid concentration slowed down around noon

probably because of fresh emissions (e.g., cooking sources) at lunchtime. $C_9$ $\omega$-oxo acid and diacids (e.g., nonanoic acid, 9-oxononanoic acid and azelaic acid) in the atmospheric environment originate from plant volatilization, combustion emissions and cooking processes (Kawamura et al., 2013; Tian et al., 2020), and they were established in chamber studies as major atmospheric oxidation products from uFAs ozonolysis (Kawamura et al., 2013; Moise and Rudich, 2002; Thornberry and Abbatt, 2004). The diurnal variations in nonanoic acid and 9-oxononanoic acid were similar, and both peaked around noon, while 9-oxononanoic acid and azelaic acid are in competition (Thornberry and Abbatt, 2004). However, the concentration of 9-oxononanoic acid was significantly higher than that of nonanoic acid (Fig. 8c and d), which may be due to the following reasons: (1) 9-oxononanoic acid can be produced by two pathways, while nonanoic acid generation can only be produced through one of the pathways competing with nonanal, and the molarity generated from the ozonolysis of oleic acid is smaller than that of 9-oxononanoic acid (Gross et al., 2009); (2) due to the high volatility of nonanoic acid, its concentration in the particle phase is much lower, and only a small portion of nonanoic acid in PM is detected by TAG (Wang and Yu, 2021).

Figure 8b to d show the relationship between the ratios of ODPs / stearic acid and oleic acid / stearic acid. In CL#2 and CL#4, the 9-oxononanoic acid / stearic acid ratio is larger than that in CL#1 and CL#3, and the azelaic acid / stearic acid ratio has the same characteristic. The nonanoic acid / stearic acid ratio is not well characterized probably because most of the nonanoic acid is present in the gas phase. Bikkina et al. (2019) found that the O/S ratio exhibited a nonlinear (power) inverse relationship with azelaic acid in remote marine aerosols. This feature was not found

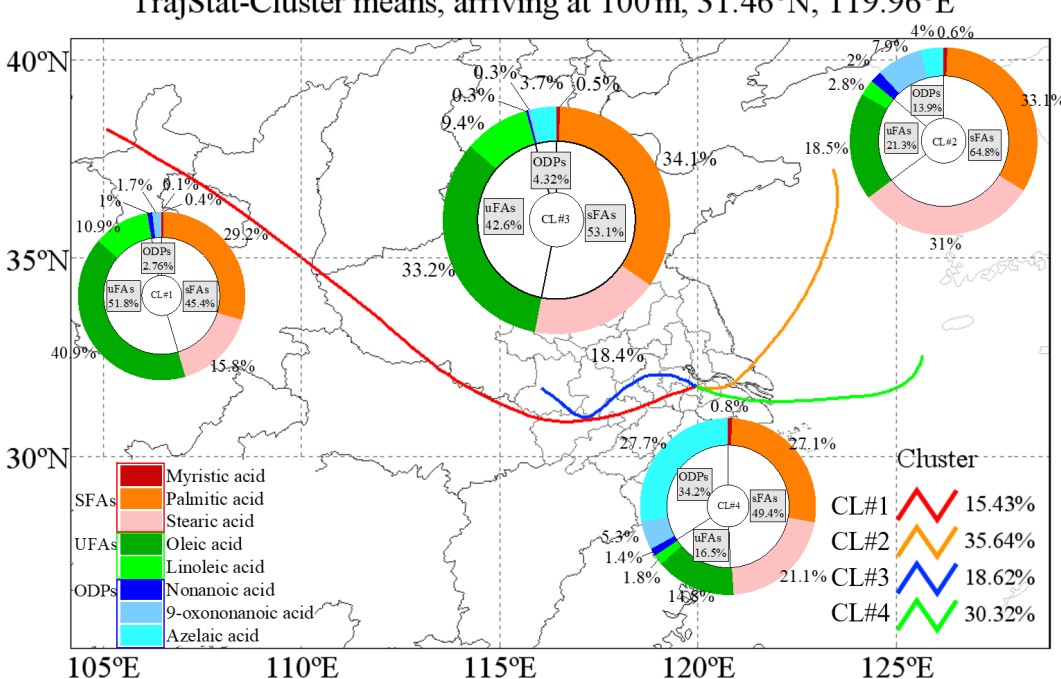

**Figure 6.** Sources for each air mass during the sampling period. The colored lines in the map show the contribution of each directional air mass source to the total trajectory as resolved by the TrajStat model.

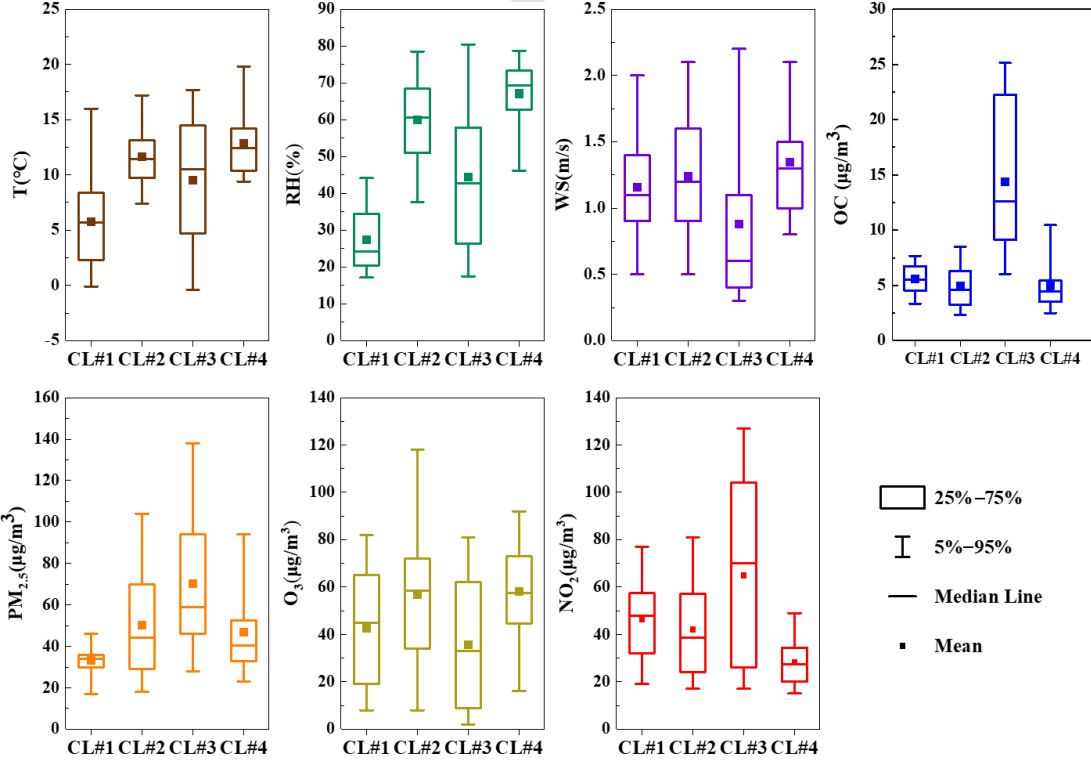

**Figure 7.** Box plots of meteorological parameters and pollutant concentrations in each cluster (squares and solid lines correspond to the mean and median, respectively; boxes indicate the 25th and 75th percentiles, and whiskers are the 5th and 95th percentiles).

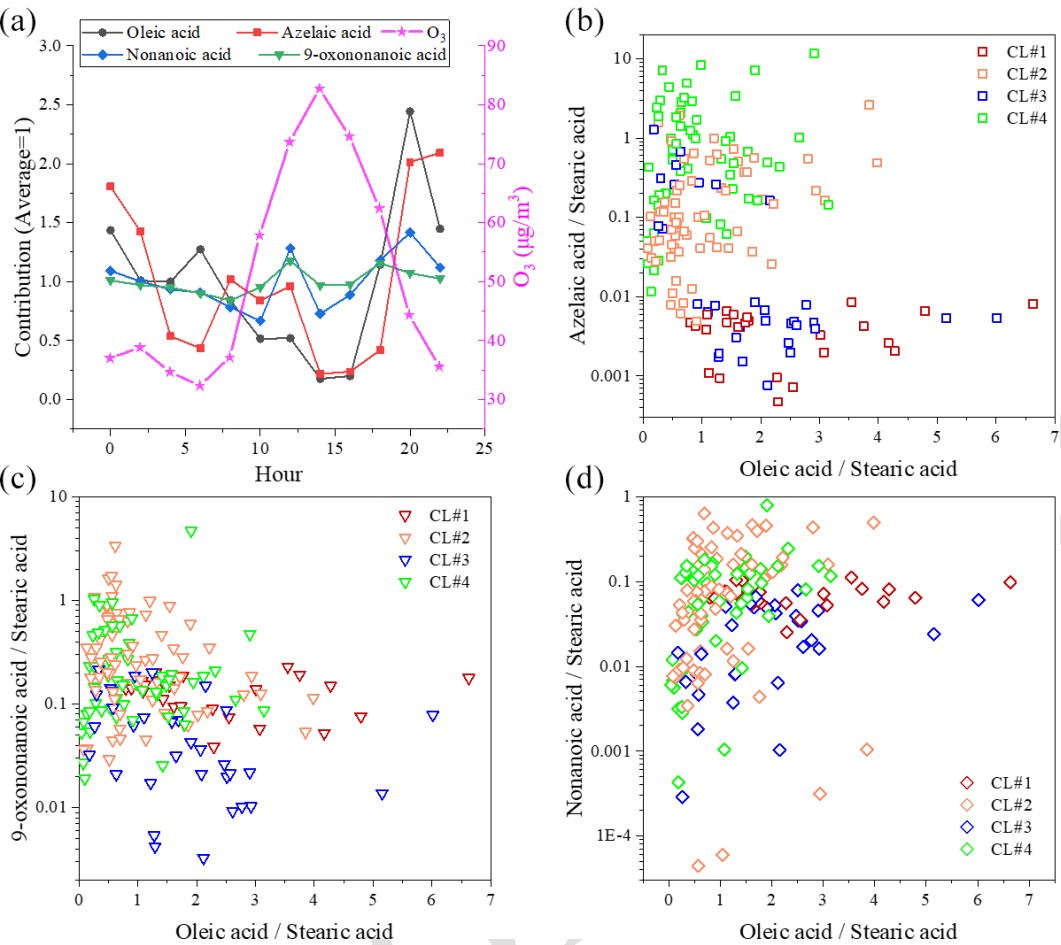

**Figure 8.** Diurnal variation in $C_9$ products and oleic acid in environmental samples compared to $O_3$ **(a)** and correlation of $C_9$ products – azelaic acid **(b)**, 9-oxononanoic acid **(c)** and nonanoic acid **(d)** – with oleic acid.

in this study, which is possibly due to the single source class of fatty acids and ODPs in remote marine areas, the diversity of emission sources in urban areas, and their vulnerability to transport.

## 3.4 Oxidative decomposition of uFAs

From the above analysis, cooking emissions were the most important source of fatty acids in atmospheric $PM_{2.5}$ in urban areas of Changzhou, especially during the dinner period. Both sFAs and uFAs peaked between 18:00 and 22:00 and then declined until breakfast time on the next day. Fatty-acid-like substances in fresh cooking emissions react with various oxidants while being continuously replenished by the fresh cooking emission during the day, so the degradation of uFAs in the particulate phase can be complicated. With no obvious fresh cooking emissions after dinner and the low volatility of the target pollutants studied (oleic and linoleic acids), the effect of gas–particle partitioning on them can be disregarded, and the evening provides a good opportunity to investigate the chemical degradation of uFAs from cooking emissions.

Therefore, we selected the period from 18:00 in the evening to 06:00 in the morning, focusing on the impact of oxidants in the atmospheric environment on uFAs. The definition of the effective rate constant $k$ has been described in previous studies (Donahue et al., 2005; Wang and Yu, 2021). To calculate the rate constant of uFAs with oxidants (especially $O_3$ and $NO_3^*$, etc.), a one-step model was utilized, and an average decay rate constant for each night could be derived. The same method has been used in the study of Wang and Yu (2021), which shows that more than 77 % of the observed data fit better with a one-step model. Figures S5 and S6 show the nighttime oxidative degradation of oleic acid and linoleic acid, respectively. It should be noted that not all of the reactants (uFAs) will be fully consumed from the start of the fit until fresh emissions enter the atmosphere, and the amount of consumed and remaining uFAs could be affected by a combination of oxidant level, source activity and meteorological conditions.

Figure 9 shows the effective rate constants of the oxidative decomposition of oleic ($k_O$) and linoleic ($k_L$) acids in relation to air oxidants ($O_3$, $NO_2$, $O_x$, $NO_3^*$, etc. $O_x$ is the

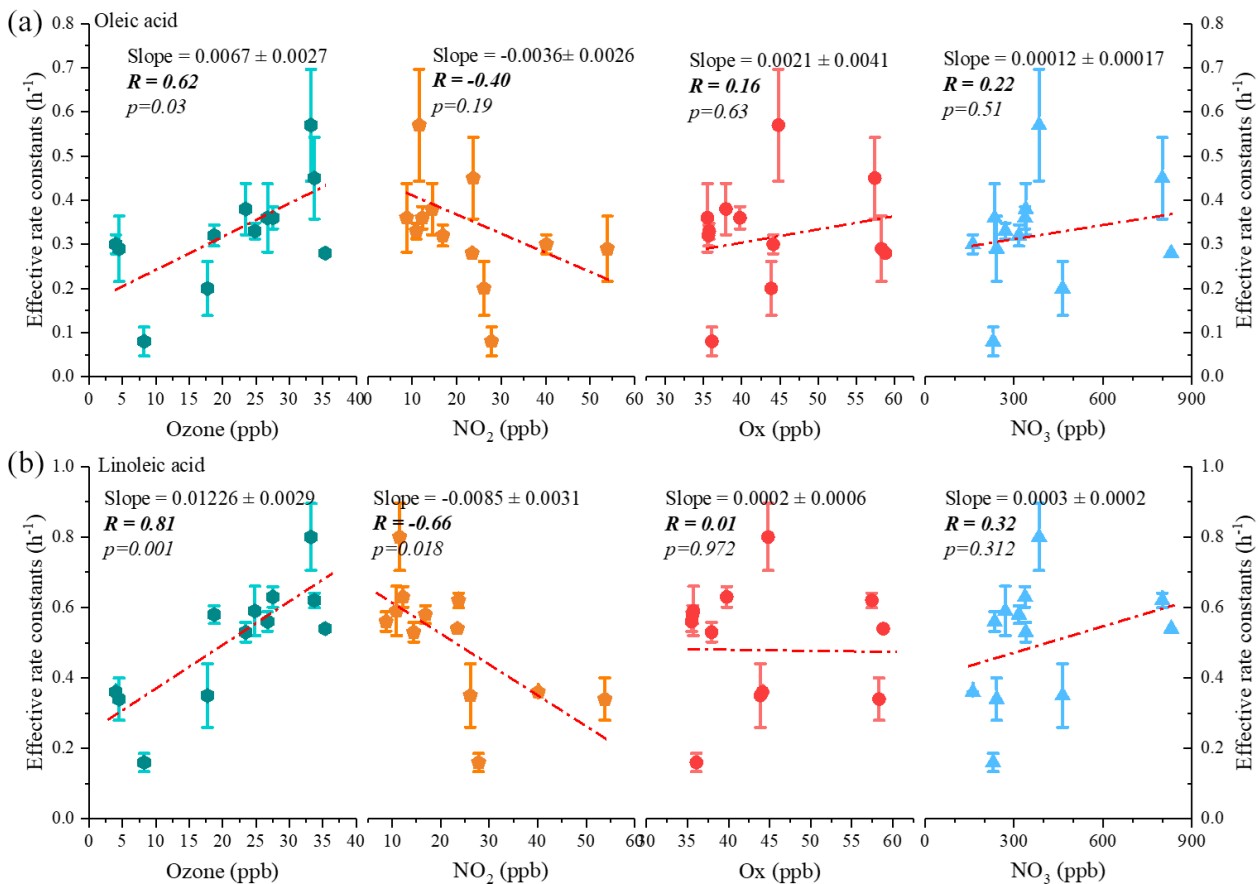

**Figure 9.** Correlations of the estimated effective decay rate constant with average nighttime atmospheric oxidant concentrations for oleic acid (**a**) and linoleic acid (**b**) (the $p$ value indicates the parameter of the $F$ test of the regression equation in the regression model).

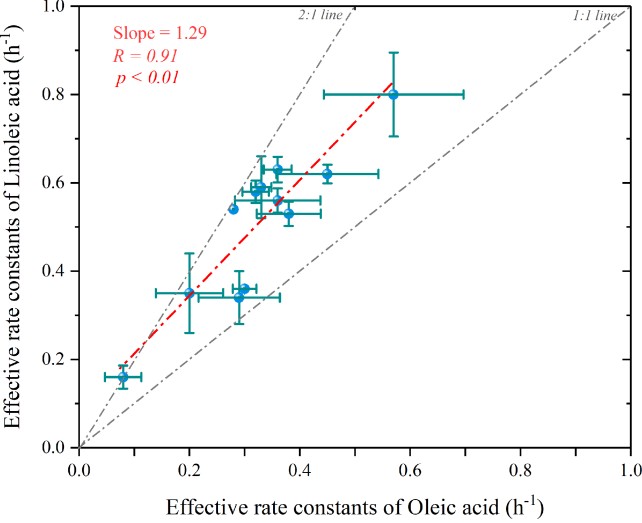

**Figure 10.** Scatterplot of the estimated effective rate constant for linoleic acid versus oleic acid (the $p$ value indicates the parameter of the $F$ test of the regression equation in the regression model).

total oxidant, calculated from $O_x = NO_2 + O_3$). It should be noted that the $NO_3^*$, calculated by multiplying $O_3$ by $NO_2$, is a substitution for the $NO_3^*$ radical, which is not available in this campaign. Both $k_O$ and $k_L$ had a significant positive correlation (the $p$ values of significance tests were all less than 0.05) with $O_3$, and no correlation was observed with other air oxidants ($O_x$, $NO_3^*$ and $NO_2$). Ozone acted as the predominant oxidant for the oxidative decomposition of uFAs, which was consistent with the conclusion in Shanghai. In addition to the oxidants mentioned above, laboratory studies have also reported $N_2O_5$ reacts with olefinic acids containing C=C bonds such as oleic acid and linoleic acid, which has a much slower reaction kinetics than that of $NO_3^*$ (Gross et al., 2009). Therefore, the effect of $N_2O_5$ was ignored in this study.

Figure 10 shows the scatter plot of the effective rate constants of oleic and linoleic acid. The significant correlation between the effective rate constants of oleic acid and linoleic acid was not equal to 1 due to the differences in aerosol composition and environmental conditions. The effective rate constant of oleic acid ranged from 0.08–0.57 $h^{-1}$, which was overall smaller than $k_L$ (0.16–0.80 $h^{-1}$), indicating that their

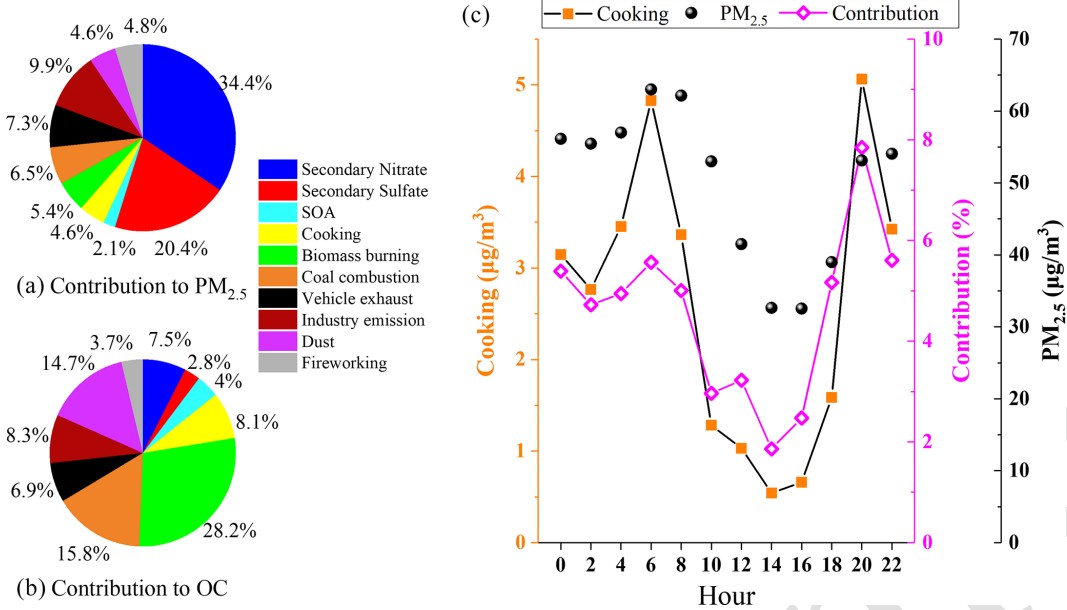

**Figure 11.** Comparison of individual factor contributions to PM$_{2.5}$ **(a)** and OC **(b)**; diurnal variation in cooking sources **(c)**.

reactivity is closely related to their chemical structure, and the two C=C bonds in the linoleic acid have a higher probability of reacting with atmospheric oxidants. However, besides the chemical structure, other factors (e.g., diffusion and temperature) also affect the calculation of the oxidation reaction rate of uFAs. The fitted ratio of $k_L/k_O$ is 1.29 (dashed red line in Fig. 11), with most scatters falling in the area with $k_L$ to $k_O$ values above the 1 : 1 ratio. $k_L/k_O$ has a mean value of $1.6 \pm 0.3$, and the relative reactivity of linoleic acid to oleic acid is below 2 in the measured environmental data but close to the results of laboratory studies with O$_3$ as oxidant. We also reviewed the $k_L/k_O$ ratios of O$_3$, NO$_3^*$ and N$_2$O$_5$ as oxidants in other laboratory studies, and the $k_L/k_O$ ratios of the three oxidants were 1.7, 1.8 and 2.9 (Gross et al., 2009; Thornberry and Abbatt, 2004), respectively. The relative reaction coefficients $k_L/k_O$ measured for O$_3$ in laboratory studies are close to our results. The comparison indicates that O$_3$ was the most likely oxidants for the nighttime uFAs oxidation in the urban area of Changzhou.

## 3.5 Source contributions of cooking aerosol to PM$_{2.5}$ and OC

To gain a more quantitative assessment of the source contribution from cooking to OA, PMF was applied for source apportionment. The target POA markers were incorporated into the input data matrix, along with secondary organic aerosol (SOA) markers (Table S1) and major aerosol components including major ions, elements, EC and OC. Source apportionment of PM$_{2.5}$ in this field campaign yielded 10 sources, including three secondary sources (secondary sulfate, secondary nitrate and SOAs) and seven primary emis-

sion sources (cooking, biomass burning, coal combustion, vehicle exhaust, industrial emissions, dust and fireworking). A detailed description of the identification of each PMF-resolved source factor is shown in Sect. S1. Briefly, secondary source factors account for the largest share of PM$_{2.5}$ (the total was 56.9 %, of which secondary nitrate contributes up to 34.4 %), and primary emissions contributed 43.1 % of total PM$_{2.5}$ (Fig. 11). Among the primary source factors, industry makes the largest contribution to PM$_{2.5}$ mass concentration (9.9 %).

In a specific polluted period, different sources have different impacts on the PM$_{2.5}$ concentration and chemical composition in Changzhou. Among the 10 sources, the cooking factor was dominated by sFAs and uFAs during the monitoring period, accounting for 4.6 % of the total PM$_{2.5}$. The concentration of cooking sources and its contribution to total PM$_{2.5}$ also showed a clear diurnal variation, with two peaks at around 06:00 and 20:00, especially at dinnertime. The contribution of cooking to PM$_{2.5}$ concentration during mealtime increased significantly compared with other periods, reaching 7.8 % at 20:00. The mean concentration of cooking aerosol in the polluted period was estimated to be $4.0\,\mu g\,m^{-3}$, which was 5.3 times higher than that in the clean period ($0.75\,\mu g\,m^{-3}$). The variation was similar to that of fatty acids. The factor profiles of the 10-factor constrained run of PMF are shown in Sect. S1 and Fig. S5 in the Supplement, together with the time series of contributions from individual source factors. Overall, we estimated that cooking accounted for 5.8 % of the total PM$_{2.5}$ during the polluted period, which was 1.9 times greater than that of 3.0 % during the clean period. During the whole observation period, the cooking factor contributes only a small part of PM$_{2.5}$ (4.6 %),

but it accounts for 8.1 % of the total OC, indicating the importance of cooking emissions to organic matter, which is a significant source of organic pollution in urban areas.

# 4 Conclusions

In this study, we measured uFAs, sFAs and ODPs every 2 h using TAG in urban Changzhou. The concentration of TFAs averaged at 105.70 ng m$^{-3}$, close to that in Shanghai. The average concentration of TFAs in the polluted period was 147.06 ng m$^{-3}$, which was 4.2 times higher than that during the clean period. During the rising period of PM$_{2.5}$, the TFA concentration tends to reach the peak earlier than PM$_{2.5}$, and the proportion of TFAs in PM$_{2.5}$, as well as OC, will increase first and then decrease. However, when affected by adverse diffusion, the TFA concentration will accumulate continuously as PM$_{2.5}$. The linoleic acid / stearic acid and oleic acid / stearic acid ratios exhibited a significant peak during dinnertime, which was close to the cooking source profile values, and a relatively smaller peak at lunchtime. Cooking sources during dinner hours are the most important contributors to the concentration of fatty acids in PM$_{2.5}$ during the study period. The diurnal trend of ODPs was different from that of uFAs, and the concentration of ODPs increased significantly at noon. The diurnal variations in nonanoic acid and 9-oxononanoic acid in ODPs are similar mainly because oleic acid can produce both 9-oxononanoic acid and nonanoic acid in the ozonolysis pathway.

Under the influence of different air masses, there were significant variations in the ratios of various organic acids from cooking. The highest total concentrations of sFAs, uFAs and ODPs were found under the local air mass cluster (CL#3), indicating significant local emissions contributing fatty acids, as well as PM$_{2.5}$. The percentages of TFAs in CL#1 and CL#3 were larger than those in CL#2 and CL#4. The proportion of ODPs in CL#2 and CL#4 was greater than that in CL#1 and CL#3. This is mainly because under the influence of transportation the air masses brought more sFAs and ODPs. The air masses were more aged, and the higher ozone concentration and more active uFAs decomposition reaction occurred in these two air mass clusters. The daily oxidative degradation kinetics of oleic and linoleic acids were obtained using data during the nighttime on each observation date. The $k_O$ ranged from 0.08 to 0.57 h$^{-1}$, which was overall smaller than $k_L$ (0.16–0.80 h$^{-1}$). It was observed that both $k_O$ and $k_L$ had a significant positive correlation with O$_3$. The relative reaction coefficients $k_L/k_O$ (1.6±0.3) of linoleic and oleic acids in this study are close to $k_L/k_O$ measured for O$_3$ in laboratory studies, indicating that O$_3$ was the main nighttime oxidant for uFAs in Changzhou. Overall, this study describes the concentration variation and oxidative degradation of uFAs and oxidation products in ambient air based on hourly time-resolved observations, guiding future refinement of source apportionment of PM$_{2.5}$ and the development of cooking emission control policies.

The average contribution of cooking to PM$_{2.5}$ was estimated to be 4.6 %, while the average contribution to total OC was 8.1 %. However, the proportion of cooking to total PM$_{2.5}$ among different sources during the meal period increased significantly compared with other periods, especially during the dinner period, peaking at 7.8 %. It is estimated that cooking sources accounted for 5.8 % of the total PM$_{2.5}$ during the polluted period, which was 1.9 times greater than the 3.0 % during the clean period, showing that more attention should be paid to strict controls on cooking emissions during pollution episodes.

**Data availability.** All data used in this paper have been deposited in an open research repository, which is available at https://doi.org/10.17632/h67km6dnxn.1 (Li, 2023)TS2.

**Supplement.** The supplement related to this article is available online at: https://doi.org/10.5194/acp-23-1-2023-supplement.TS3

**Author contributions.** RL, KZ, QL and LY conducted the field measurements. RL and KZ performed the data analysis and prepared the manuscript with contributions from all co-authors. LL formulated the research goals and edited and reviewed the manuscript. LL and JZY reviewed and edited the manuscript. All authors contributed to data interpretations and discussions.

**Competing interests.** The contact author has declared that none of the authors has any competing interests.

**Acknowledgements.** This study is financially supported by the National Natural Science Foundation of China (nos. 41875161, 42075144 and 42005112). We thank the Changzhou Environmental Monitoring Center of Jiangsu Province for their help in conducting the field campaign.

**Financial support.** This research has been supported by the National Natural Science Foundation of China (grant nos. 41875161, 42075144 and 42005112).

**Review statement.** This paper was edited by James Allan and reviewed by three anonymous referees.

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

## Remarks from the typesetter

**TS1** We have not adjusted the value here. Meaning and content changes, including changes to values, should be reviewed by the editor before being implemented in the proofreading stage. Please reassess if these changes are strictly necessary before taking this step. For more information, please see our proofreading guidelines at: http://publications.copernicus.org/for_authors/proofreading_guidelines.html. If you want us to change the values, please prepare an explanatory document (doc or pdf) which we can send to the editor via our system.