# Peer review of "Characteristics and degradation of organic aerosols from cooking"

_Atmospheric Chemistry and Physics, 2022_

## Referee Comment (RC2)

Review of
**"Characteristics and degradation of organic aerosols from cooking sources based on hourly observation of organic molecular markers in urban environment"**
By Li et al.

General comments

This study aims to characterise the cooking organic aerosol markers evolution in the atmosphere. The study employs online thermal desorption aerosol gas chromatography mass spectrometry (TAG) to analyse the organic component of PM2.5 from urban Changzhou, China. The study identified and attributed saturated fatty acids, unsaturated fatty acids, and oxidative decomposition products of unsaturated fatty acids as the molecular marker of cooking sources. Additionally, the decomposition of these markers was estimated using an established ageing parameterisation model, and the contribution of the cooking source to PM2.5 was estimated using a positive matrix factorisation model.

The online molecular analysis employed by this study gives a great deal of data that, unfortunately not being optimally used in the analysis and discussion. The study adopted and modified analysis methods by Wang et al. (2020) and Wang and Yu (2021). However, the modification is not entirely justified, raising more questions about the results and discussion. Finally, the association of fatty acids as cooking source molecular markers need more evidence/analysis that considers the other sources, such as biomass burning and marine aerosol.

The manuscript is well written, with some typos that can be improved after thorough checking. The topic presented in this study fits within the scope of the *Atmospheric Chemistry and Physics* journal. My main concerns are the insufficient evidence to support the cooking organic aerosol molecular marker characterisation and their atmospheric decomposition. In summary, I recommend the Editor reconsider the manuscript after major revisions.

Specific comments

1.  Although following previous studies, The Methods section needs explanation and clarifications, which are as listed below.
    a.  The PM2.5 sampling method (Lines 94-96: flow rate, residence time, type of collection matrix, etc)
    b.  The post-sampling analysis (Lines 100-102: brief procedures). For example, an hourly sampling will result in 24 samples at maximum. An hour of sampling plus 1.5-hour post processing before starting a new sampling sequence will result in less than 12 samples per day.
2.  The identification of cooking aerosol markers needs improvement. Attributing all Fatty Acids as cooking aerosol markers is misleading because they are also emitted by other sources, i.e., biomass burning and traffic. The study used a similar measurement method (TAG) to Wang et al. (2020) and is referred to it in the Methods section. That means this study measured other organic molecules, such as sugars (levoglucosan) and PAHs, similar to Wang et al. (2020) study. Moreover, the Supplement Section S2 Figure S7a shows source

profiles composed of small and long-chain acids, PAHs, and sugars, suggesting that these data are available for discussion. These other organic groups would improve the identification of cooking source contribution to the total fatty acids emissions.

    a. Lines 71-74: They are not only markers for cooking/culinary emissions. They can also be emitted by biomass burning, vehicles, and plants (Rogge et al., 1991, 1993, 1998; Simoneit, 2002). Since there are a couple of potential emission sources, discuss how previous studies differentiate them (Ho et al., 2015, Wang et al., 2020). Ref: Simoneit (2002, https://doi.org/10.1016/S0883-2927(01)00061-0), Rogge et al. (1998, https://doi.org/10.1021/es960930b), Rogge et al. (1991, https://doi.org/10.1021/es00018a015), Rogge et al. (1993, https://doi.org/10.1021/es00041a007), Ho et al. (2015, doi:10.5194/acp-15-3111-2015)

    b. What are EP1, EP2, etc., in Figure 2. A summary table of EP1, EP2, etc., parameters can be added to the Supplement. Additionally, plotting fatty acids in log scale make the time trends incomparable to the other parameters plotted in linear scale (Figure 2). What are the reasons behind separating EP2 and EP3 (Figures 2 and 4)? Instead of grouping and averaging the measurement into day-measurement (D1, D2, D3, etc.), time series of TFAs/OC for EP1, EP2, etc. would show the actual trend and give better insight into the transformation of the fatty acids (Lines 265-269).

    c. Section 3.1: How TFAs and FAs are differentiated here? TFAs (Line 168) are similar to FAs (Line 176). Be consistent with using fatty acids or FAs, and FAs or TFAs. Additionally, Considering the other sources of fatty acids, the missing of lunchtime raises a question (Lines 180-185). Could the fatty acids come from biomass burning (residential heating) at night and vehicle emissions in the morning? Wang et al. (2020) observed a small increase in the daytime. Additionally, back trajectory analysis suggests CL#2 and CL#4 airmasses came from the sea (Line 223, Figure 5), suggesting a marine aerosol potential contribution to fatty acids.

    d. Based on Figures 4 and 3, the cooking source seems to contribute at night only. In the morning, the fatty acids could be contributed from non-cooking or long-range transport sources, of which the latter can be a combination of sources. Additionally, Oleic/Palmitic in Figure 4 is peaking at noontime, which could be associated with fresh cooking emissions at lunchtime. Instead of individual species analysis, cluster analysis of the ratios of the fatty acids could give further insight into their atmospheric evolution.

3. This study used parameterisation by Wang and Yu (2021) to estimate the fatty acids atmospheric ageing. Wang and Yu (2021) developed the relative rate constant for the ambient fatty acids based on C18:0 stearic acid as the reference molecule for normalisation.

    a. Any reason for choosing Palmitic Acid instead of Stearic Acid (Lines 132-133)? Wang 2020 plotted Oleic/Stearic or Linoleic/Stearic vs Palmitic/Stearic to

investigate the ageing of cooking markers. Is there a particular reason for plotting Oleic/Palmitic or Linoleic/Palmitic versus Palmitic/Stearic (Lines 194-202, Figure 4)? Using different denominators would result in different ageing interpretations. However, it could be good to test the effect of Stearic Acid and Palmitic Acid as the ageing reference to the analysis.

    b. Lines 292-293, Figure 8: The plot does not show negative linear correlation as inferred in the text and figure. The curves could be potentially negative exponential for Azelaic or Oxononanoic/Palmitic vs Oleic/Palmitic, but not linear. Moreover, there is no R or $R^2$ value for the regression plots. The R or $R^2$ value is important to assess the association between the two ratios/parameters. Another method is calculating the p-value to show the statistical significance of the two ratios.

    c. What is the reason for plotting Figure 9 in a 2-power scale instead of a linear scale? If these axes are correlation values for X9/P vs O/P, Figure 9 should plot $R^2$ values of each ratio.

4. This study used PMF analysis to estimate cooking source contribution to PM2.5. However, this study has not adequately discussed the PMF analysis leading to the conclusion of cooking source contribution. It is mentioned that this study would only briefly identify each source factor. However, a discussion of the PMF analysis is still needed to support the conclusion on cooking aerosol contribution to PM2.5. The PMF analysis can go to the Supplement as done in this manuscript, but it needs more information. The authors should provide, at a minimum, the correlation coefficient of factor solution profiles and time series and the observed and predicted cooking aerosol markers. Further information, such as the Base Model Displacement Error Method and Bootstrapping analysis, is also important to explore the rotational ambiguity and assess the uncertainty that arises from random errors in the dataset, respectively (e.g., Almeida et al. (2020, https://doi.org/10.1016/j.envpol.2020.115199). Lastly, considering the cooking contribution is smaller than biomass burning, and vehicle exhaust (Figure 12), fatty acids discussed here might not be mainly emitted by cooking activities.

Technical comments

a) Lines 227-228: The component of ODPs haven't been explained in the text.

b) Figure 5: Add a legend to explain the coloured lines for the Cluster.

c) Lines 260-262: Add reference studies or ratios for TFAs/OC from other sources.

d) Figure 8a: What does the y-axis Contribution mean?

e) Line 311: Correlation as in correlation coefficient values or relationship between Y and X axes?

f) Lines 391-302, Figure S7: There is no time series or diurnal plot of the PMF factor solution in the Supplement.

---

## Author Comment (AC1)

**Response to comments by Reviewer#1**

We thank the reviewer for the detailed and constructive comments and suggestions. Below is our point-by-point response to each comment, marked in blue. Changes made to the main text are marked in red. Revisions made to the manuscript are highlighted.

**General comments:**

This manuscript presents measurements of fatty acids and their oxidation products, which are used as a proxy for cooking emissions, at an urban site in Changzhou, China. Aerosol gas chromotography-mass spectrometry (TAG) was used to detect concentrations of these compounds every two hours during three monitoring periods in January – March 2021. The authors explore changes in concentration throughout the diurnal cycle and under the influence of different air masses. Estimates of the effective rate constant are used to suggest that the key night-time oxidant for these fatty acids is ozone.

This paper's specific focus on cooking emissions provides a valuable contribution towards a growing literature base on the chemical composition of urban aerosol. The overall goal of the paper – to quantify the contribution of cooking aerosol and examine its influence under different regimes – is important. However, I found the conclusions in some parts of the paper to be unclear or not fully supported by the data presented. I would also recommend a check through the paper for typos and to improve the clarity of the writing in some places. I would suggest that this paper is suitable for publication in ACP, subject to some edits. Additional comments are provided below

**Response:** We thank the reviewer for the positive comments. We have carefully checked the manuscript to make sure that the conclusions are clear and could be well supported by the data presented. We have also checked and revised the typos throughout the paper. Revisions are highlighted.

**Major comments in detail:**

1. Line 95: This would benefit from more information about the process of how sample collection takes place with TAG.

**Response:** We have added descriptions about sampling to the main text. Please refer to Lines 93-

110.

**Lines 93-110**: "Quantification of hourly speciated organic markers was achieved using TAG. The operation details and data quality have been described in our previous work (Wang et al., 2020; Zhang et al., 2021). The sampling and analysis sequence of the TAG system includes four steps: (a) $PM_{2.5}$ sampling and synchronous gas chromatography-mass spectrometry (GC-MS) analysis of the previous sample; (b) loading of the internal standards (IS) from the standards (STD) reservoir to a thermal desorption cell; (c) derivatization and thermal desorption of analytes on the collection and thermal desorption (CTD) cell and subsequent preconcentration of the analytes in focusing trap (FT); and (d) loading of analytes into the GC column for GC-MS analysis. The following is a detailed description. Ambient air was sampled at a flow rate of 8.5-9.5 L/min through a cyclone with $PM_{2.5}$ cutting size (BGI Inc., Waltham, MA), a Nafion dryer (PERMA PURE, MD-700-24S-3) to remove moisture, and then through a carbon denuder (model: ADI-DEN2) to remove volatile organics. The sampled particles were collected on the CTD cell at 30°C for 60 min, followed by derivatization and thermal desorption for 8 min as the temperature of the CTD cell increases to 300°C in 2 min and maintains for 6 min, during which a 10 mL/min helium purge flow combined with a 40 mL/min derivatization flow with N-methyl- N-(trimethylsilyl) trifluoroacetamide (MSTFA) flow through for 8 min. Subsequently, the FT was heated to 300°C in 2 min and kept at 300°C for 10 min, transferring the analytes onto the GC column head (DB-5MS, size 30 m × 0.25 μm × 0.25 μm) by carrier gas. After GC separation, the target organics were sent to the MS detector for quantification. The GC-MS analysis duration for each sample was 60 min while collection of the next sample the CTD cell starts. With the current TAG instrumental set-up, samples were collected every even hour. The post-sampling steps, including in-situ derivatization, thermal desorption, GC-MS analysis, and standby step, took 2 h, thus producing 12 samples per day."

2. Lines 181-184 and Fig. 3: The authors discuss the influence of boundary layer changes on the concentration of fatty acids, suggesting that this is the reason there is no lunchtime increase in fatty acids. I agree that the diurnal pattern shown in Fig. 3 is very characteristic of a parameter influenced by boundary layer dynamics. However, this makes it difficult to discern which changes are related to boundary layer changing and which to fresh emissions. I might suggest that plotting the diurnal of each compound as a fraction of $PM_{2.5}$ or OC would give more insight into the chemical changes.

In fact, this can be seen later in Fig. 12, where there is a more obvious midday peak in the fractional contribution of the cooking factor than in the absolute concentrations.

**Response:** Thanks for the constructive comment. We have revised the figure to include diurnal trends of TFAs/OC and inserted relative discussions as well, please refer to Lines 205-211.

**Lines 205-211:** "Fatty acids showed a clear diurnal variation, with two peaks observed at around 6:00 and 20:00 local time, respectively, and the dinner time peak was especially prominent. In contrast to the previous observations in Shanghai, no peak was observed at lunchtime. The relatively higher boundary layer during the daytime, facilitated the diffusion of pollutants. The weaker oxidation of uFAs emitted at night made the fatty acid concentration peaks more pronounced at dinner time (Wang et al., 2020). Figure 3(b) shows the contribution of various fatty acids to OC. When the influence of the boundary layer height change was eliminated, the proportion of the five fatty acids and TFAs in OC at noon had a weaker peak, which was still smaller than that during the morning and evening mealtimes."

3. The authors assume that all fatty acids observed are the result of cooking emissions. This may be a reasonable assumption in this environment; however, I would like to see more justification for this and discussion of other potential sources. For example, palmitic, stearic and oleic acids can all be associated with biomass burning (eg Bertrand et al., 2018; Fujii et al., 2015); and palmitic and stearic acids are released from marine biota in the ocean surface (eg Bikkina et al., 2019), which could be an influence here in CL#2 and CL#4. It seems to me that fatty acids being released at the ocean surface and then oxidised during transport is a more likely explanation for the chemical composition in CL#4.

**Response:** Thanks for the constructive comment. We have reviewed the molecular marker fingerprinting of fatty acids from different sources (palmitic acid to stearic acid ratio, P/S), and the relationship between P/S and the ratio of oleic acid /stearic acid (O/S) in organic molecular markers emitted from different cooking types (Fig.4). We also examined the profiles of P/S and O/S in fatty acids from sea spray aerosol emissions, and concluded that sea spray aerosol was not a major contributor to fatty acids in urban areas in this study. We could not exclude the impact of aging sea spray aerosol on non-coastal cities. Please refer to Lines 216-231.

**Lines 216-231:** "Fatty acids in urban atmospheres are influenced by various anthropogenic (e.g.,

biomass burning, vehicle exhaust) (Hays et al., 2002; Schauer et al., 2001; Simoneit, 2002; Wang et al., 2009) and biogenic sources (Oliveira et al., 2007; Rogge et al., 2006). The main sources of fatty acid-like substances in the atmospheric environment of the study area can be discerned on the basis of characteristic ratios between fatty acids emitted from different sources (Fig.4) (He et al., 2004; Pei et al., 2016; Rogge et al., 1993; Zhao et al., 2015; Zhao et al., 2007). The palmitic acid to stearic acid (P/S) ratios observed in this study had a range between 0.49 and 3.08 (average value: 1.49), significantly lower than those associated with residential coal combustion and industrial coal combustion, while partially overlapping those from biomass burning, vehicle exhaust and sea spray aerosol (Bikkin et al., 2019; Cai et al., 2017; Ho et al., 2015; Zhang et al., 2008; Zhang et al., 2007). Ho et al. (2015) studied urban areas in Beijing where fatty acid concentrations were elevated during traffic restrictions compared to non-restricted periods, suggesting that motor vehicle exhaust is not a significant source of fatty acids in urban areas. In the study of Simoneit (2002), no oleic acid was detected in organic molecular substances from biomass burning. The oleic acid/stearic acid (O/S) ratio from sea spray aerosol samples is 0.16 (Bikkin et al., 2019), which is obviously lower than the ambient data in this study (1.4). Thus, it is reasonable to conclude that biomass burning, vehicle exhaust and sea spray were insignificant sources of fatty acids in urban Changzhou during the observation in this study. Especially during the dinner period, when the O/S ratio was significantly higher and close to the ratio in the organics emitted from traditional culinary types in the Yangtze River Delta region."

4. Lines 190-193 and Fig. 4a: If I have understood correctly, palmitic acid/stearic acid (on the x-axis) is taken to be a rough indicator of the source of the emissions. If this is the case, it is difficult for the reader to interpret the x-axis here. Could it be more illustrative to label the ratios for a selection of sources on the x-axis? Would you expect to see a relationship between the y-axis and the x-axis?

**Response:** Palmitic acid (P) and Stearic acid (S) are both saturated FA. Their degradation is much slower than that of oleic acid or linoleic acid. Due to their similar chemical reactivity and volatility and cooking being their common source, the ratio P/S was expected to be relatively constant among different samples. The narrow variation range (ca 0.5-3) shown by the Fig. 4a (now Figure 5a in the revised manuscript) confirmed this point. The main point of Fig. 4a is to contrast the large variation

of the oleic/palmitic acid ratio (0.1-10) against the small variation in the P/S ratio, thus highlighting the degradation of oleic acid. Thus, no relationship was expected between the y-axis and x-axis. We now have modified the presentation of this plot by adding an data point (located at the upper right corner) indicating the average ratio of oleic/stearic and P/S ratios for fresh cooking emission. It should also be noted that in the revised manuscript, palmitic acid was replaced with stearic acid as a reference. Please refer to Lines 239-253.

**Lines 239-253:** "Therefore, the ratio of P/S mainly depends on the sources. Fig.5 shows the O/S ratios and linoleic acid/ stearic acid (L/S) versus P/S, respectively. The average value of P/S was 1.49±0.49, which was within the range of cooking source profile values measured from direct emissions from different restaurants and cooking types (1.3-8.1) (He et al., 2004; Pei et al., 2016; Schauer et al., 2002; Zhao et al., 2007), and similar to the ratio of P/S in atmospheric $PM_{2.5}$ in Shanghai (1.9) (Li et al., 2020; Wang et al., 2020). In this study, the O/S ratio (1.4 ± 1.1) of the ambient samples was overall in the range of the cooking source profile (1.2-6.5, with an average of 3.6), while the L/S ratio of 0.25 ± 0.31 was slightly lower than the cooking source profile values (1.1-5.8, and the average was 2.9) (He et al., 2004; Pei et al., 2016; Schauer et al., 2002; Zhao et al., 2007), indicating that linoleic acid is more easily degraded than oleic acid. The O/S ratio of the ambient samples in this study was higher than those measured in Beijing (0.65) (He et al., 2004) from January to October and in Shanghai (0.83) (Li et al., 2020; Wang et al., 2020) during winter.

The diurnal variations of O/S and L/S are also shown in Fig.5. The ratios were significantly higher during dinner time (18:00-20:00), and were closer to the cooking source profile. Demonstrating that fresh emissions entered into the atmosphere during cooking period, especially dinner time, while uFAs were quickly consumed during aging. The ratio of linoleic acid to stearic acid is consistently lower than what is involved in the source spectrum, which may be influenced by different regions and source characteristics from different types of restaurants."

[Figure]

Fig.5. The oleic/ stearic acid and linoleic/ stearic acid ratios compared to the palmitic/stearic acid ratio (a); diurnal variation in the ratio of oleic (linoleic) acid to stearic acid concentration (b).

5. Section 3.3 and Fig. 7: From Fig. 7, there does not appear to be a relationship at all between the concentration of fatty acids and that of $PM_{2.5}$, which makes it difficult to discern the key scientific conclusion from this analysis. In the text, the authors point to a link between the air mass cluster and the behavior of TFAs vs $PM_{2.5}$. I would suggest clarifying the main conclusion from this section and ensuring that it is clearly supported by the figure.

**Responses:** Thanks for pointing this out. After a careful re-analysis of the data and figures, we agree that no clear scientific conclusions could be drawn from the original Fig 7. Therefore, we have reorganized this in the manuscript. Specifically, the first half of section 3.3 (Original lines 248-263 and Table 2) was retained and moved to subsection 3.1 (New manuscript, lines 183-201 and table 2).

6. Fig. 8: The correlations shown here appear to be quite weak; conclusions taken from them would be strengthened with further statistical analysis. I might suggest:

    a. Considering a log-log axis,

    b. Carrying out a t-test in each case to establish whether the correlations are statistically significant,

    c. Stating correlation co-efficients in the plots.

7. The purpose of the correlation plots (Fig. 8b-d) is not clearly explained in the text.

**Responses:** Thanks for the comments. We adjusted the section "Atmospheric aging of cooking

markers" by merging the original Fig 8 - Fig 9 and removing Fig.8. The logarithmic form is used for the y-axis to better distinguish the differences in ODPs/Stearic acid under different air mass clusters. Although no correlation analysis was conducted in Fig.8 such that a t-test was not performed, we agree with the reviewer to use significance analysis in other sections of this manuscript to make the results more convincing. Please refer to Lines 299-302.

**Lines 299-302:** "Fig.8(a) shows the diurnal variation of ozone, oleic acid, and ODPs. The ozone concentration started to rise in the morning (06:00) and peaked in the late afternoon (14:00). The diurnal trend of oleic acid was opposite to that of ozone. The diurnal trend of ODPs was also different from oleic acid, the small peak of ODPs was found at around 12:00 in the daytime, which was earlier than that of ozone."

**Lines 316-322:** "Fig.8(b) to (d) show the relationship between ODPs / stearic acid ratio and oleic acid/stearic acid. In CL#2 and CL#4, 9-oxononanoic acid / stearic acid ratio is larger than that in CL#1 and CL#3, and azelaic acid /stearic acid ratio have the same characteristic. The nonanoic acid / stearic acid ratio is not well characterized, probably because most of the nonanoic acid is present in the gas phase. Bikkina et al. (2019) found that the O/S ratio exhibited a nonlinear (power) inverse relationship with azelaic acid in remote marine aerosols. This feature was not found in this study, which is possibly due to the single source class of fatty acids and ODPs in remote marine areas, the diversity of emission sources in urban areas, and their vulnerability to transport."

[Figure]

**(new) Fig.8.** Diurnal variation of C9 products and oleic acid in environmental samples compared to O3 (a); Correlation of C9 products azelaic acid (b), 9-oxononanoic acid (c), and nonanoic acid

(d) with oleic acid.

**Minor comments**

1. Line 30: The meaning of the term 'rising period' is unclear

**Response:** We have revised the manuscript and made corrections to the abstract. "During the rising period of PM$_{2.5}$·····") was removed.

2. Line 56-8: Quantifying the concentration of fatty acids will not reduce their impact, though it may inform policy. The phrasing needs to be reconsidered here.

**Response:** We have revised the relevant description, please refer to Lines 56-59:

**Lines 56-59**, "By clarifying the characteristics of cooking emissions, quantifying the concentrations of pollutants emitted from cooking and its contribution to urban OA on the diurnal time scales, we build up data and process knowledge about cooking-sourced PM$_{2.5}$ pollution, which in turn help us evaluate the option of controlling cooking emissions in the overall pollution prevention for urban environments."

3. Sometimes 'fatty acids' is written out fully (eg line 176), and sometimes it's abbreviated to 'FA' (line 174). I would recommend using the same approach each time.

**Response:** We have checked the entire manuscript, and changed all the 'FA' to 'fatty acids' consistently.

4. Fig. 4b: I don't completely follow where the source profiles come from here. It would be good to outline the origin of these more clearly in the text.

**Response:** Thanks for pointing this out. We have inserted detailed descriptions to make it clear, please refer to Lines 216-231:

**Lines 216-231:** "Fatty acids in urban atmospheres are influenced by various anthropogenic (e.g., biomass burning, vehicle exhaust) (Hays et al., 2002; Schauer et al., 2001; Simoneit, 2002; Wang et al., 2009) and biogenic sources (Oliveira et al., 2007; Rogge et al., 2006). The main sources of fatty acid-like substances in the atmospheric environment of the study area can be discerned on the basis of characteristic ratios between fatty acids emitted from different sources (Fig.4) (He et al., 2004;

Pei et al., 2016; Rogge et al., 1993; Zhao et al., 2015; Zhao et al., 2007). The palmitic acid to stearic acid (P/S) ratios observed in this study had a range between 0.49 and 3.08 (average value: 1.49), significantly lower than those associated with residential coal combustion and industrial coal combustion, while partially overlapping those from biomass burning, vehicle exhaust and sea spray aerosol (Bikkin et al., 2019; Cai et al., 2017; Ho et al., 2015; Zhang et al., 2008; Zhang et al., 2007). Ho et al. (2015) studied urban areas in Beijing where fatty acid concentrations were elevated during traffic restrictions compared to non-restricted periods, suggesting that motor vehicle exhaust is not a significant source of fatty acids in urban areas. In the study of Simoneit (2002), no oleic acid was detected in organic molecular substances from biomass burning. The oleic acid/stearic acid (O/S) ratio from sea spray aerosol samples is 0.16 (Bikkin et al., 2019), which is obviously lower than the ambient data in this study (1.4). Thus, it is reasonable to conclude that biomass burning, vehicle exhaust and sea spray were insignificant sources of fatty acids in urban Changzhou during the observation in this study. Especially during the dinner period, when the O/S ratio was significantly higher and close to the ratio in the organics emitted from traditional culinary types in the Yangtze River Delta region."

5. Fig. 5: This figure shows the chemical influences of the different clusters very clearly. However, I would recommend rearranging slightly so that the pie charts do not block the back trajectories, and so that the text is slightly larger in some places. It is currently very difficult to read the percentages on CL#1 and CL#4.

**Response:** Thanks for the suggestion, the figure has been revised accordingly.

[Figure]

**Fig. 6** Sources for each air mass during the sampling period. The colored lines in the map show the contribution of each directional air mass source to the total trajectory as resolved by the TrajStat model.

6. Lines 235-239: I might suggest rephrasing both of these reasons, as they are currently quite difficult to understand. What is meant by the "CL#4 air mass [being] under the influence of CL#4", for example?

**Response:** Thanks for the questions and suggestions, we have corrected the description in the manuscript.

**Lines 284-290:** "The value of uFAs /sFAs in CL#2 and CL#4 was less than that in CL#1 and CL#3 and less than the ratio in sources. In addition, the proportion of ODPs in CL#2 and CL#4 is greater than that in CL#1 and CL#3. This result may be explained by the following two reasons: first, under the influence of transport, the air masses brought more sFAs, ODPs, and the air masses were more aged; second, under the influence of CL#2 and CL#4 air masses, in which the ozone concentration was higher than other air masses, the decomposition reaction of uFAs was more active and could produce more ODPs. In addition, the oxidative reaction of uFAs could be influenced by meteorological conditions as well."

7. Lines 281-2: It could be helpful to remind the reader what the conclusion referenced here was.

**Response:** Please refer to the response to comment 5, where we have revised section 3 of the original manuscript. The related paragraph has been removed.

8. Line 351: What does Ox refer to, in this context? How was it measured/observed?

**Response:** $O_x$ refers to the total oxidant, which is usually used to indicate the atmospheric oxidative capacity and can be calculated by summing $NO_2$ and $O_3$ (Dai. et al., 2019; Zhao. et al., 2020; Fu. Et al., 2020).

**Line 341:** $O_x$ is the total oxidant, calculated from $Ox = NO_2 + O_3$.

9. Fig. 12: It's difficult to interpret the relative size of sections on a 3D pie chart. I would strongly advise making this 2D.

**Response:** The figure has been revised accordingly.

[Figure]

**Fig.11** Comparison of individual factor contributions to $PM_{2.5}$ (a) and OC (b); diurnal variation of cooking source (c).

10. Lines 416-8: The authors state what was observed; it would be good to state the conclusions drawn from these observations as well.

**Response:** Corrections have been made in the manuscript, please refer to Lines 404-408.

**Lines 404-408:** "The linoleic acid /stearic acid and oleic acid /stearic acid ratios exhibited a significant peak during dinnertime, which was close to the cooking source profile values, and a

relatively smaller peak at lunchtime. Cooking sources during dinner hours are the most important contributors to the concentration of fatty acids in $PM_{2.5}$ during the study period. Diurnal trend of ODPs was different from that of uFAs, and the concentration of ODPs increased significantly at noon."

11. Line 422: It would help here to remind the reader that CL#3 was the slower-moving, more local cluster. The same goes for the other clusters when they are mentioned in the conclusion

**Response:** Thanks for the comment. Revisions have been made, please refer to Lines 279-284.

**Lines 279 -284:** "CL#3 was a slowly moving, local cluster. Under this air mass clustering, local emissions contribute significantly to fatty acids as well as $PM_{2.5}$ concentration. The air mass of CL#1 exhibits the longest range, the concentrations of ODPs were relatively small among all air masses, and the low ODPs concentration was inconsistent with other literature findings of more aging aerosol production from long-range transport (Wang et al., 2020). The lowest $PM_{2.5}$ concentrations and cleaner air masses during air mass CL#1 suggested that long-range air mass transport from the northwest was not the main source of fatty acids and ODPs in Changzhou during the observation."

**Lines 412-413:** "Highest total concentrations of sFAs, uFAs and ODPs were found under the local air mass cluster (CL#3), indicating significant local emissions contributing to fatty acids as well as $PM_{2.5}$."

12. Line 425-6: Why would this be the case? Is it because these clusters provide more ozone?

**Response:** Please refer to response to comment 6, we have modified the discussion. See Lines 285-290 and Lines 413-417.

**Lines 285-290:** "In addition, the proportion of ODPs in CL#2 and CL#4 is greater than that in CL#1 and CL#3. This result may be explained by the following two reasons: first, under the influence of transport, the air masses brought more sFAs, ODPs, and the air masses were more aged; second, under the influence of CL#2 and CL#4 air masses, in which the ozone concentration was higher than other air masses, the decomposition reaction of uFAs was more active and could produce more ODPs. In addition, the oxidative reaction of uFAs could be influenced by meteorological conditions as well."

**Lines 413-417:** "And the percentages of TFAs in CL#1 and CL#3 were larger than that in CL#2 and CL#4. The proportion of ODPs in CL#2 and CL#4 was greater than that in CL#1 and CL#3. This is mainly because under the influence of transportation, the air masses brought more sFAs, ODPs. The air masses were more aged, and the higher ozone concentration and more active uFAs decomposition reaction occurred in these two air mass clusters."

13. Line 430-1: The statement here, that "$O_3$ and $NO_3$ are the main nighttime oxidants for uFAs in Changzhou City" contradicts the statement in Lines 373-4 that "$O_3$ was the most likely oxidant for the nighttime uFAs oxidation in the urban area of Changzhou"

**Response:** Thanks for pointing this out. The description has been revised in the text and in the conclusion. Please refer to Lines 340-345.

**Lines 340-345:** "Fig.9 shows the effective rate constants of the oxidative decomposition of oleic ($k_O$) and linoleic ($k_L$) acids in relation to air oxidants ($O_3$, $NO_2$, $O_x$ and $NO_3^*$, etc. Ox is the total oxidant, calculated from $O_x = NO_2 + O_3$.). It should be noted that the $NO_3^*$, calculated by multiplying $O_3$ by $NO_2$, is a substitution for $NO_3^*$ radical, which is not available in this campaign. Both $k_O$ and $k_L$ had a significant positive correlation (The P values of significance tests were all less than 0.05) with $O_3$, and no correlation was observed with other air oxidants ($O_x$, $NO_3^*$ and $NO_2$). Ozone acted as the predominant oxidant for the oxidative decomposition of uFAs, which was consistent with the conclusion in Shanghai."

**Lines 418-421:** "The $k_O$ ranged from 0.08 to 0.57 $h^{-1}$, which was overall smaller than $k_L$ (0.16-0.80 $h^{-1}$). It was observed that both $k_O$ and $k_L$ had a significant positive correlation with $O_3$. The relative reaction coefficients $k_L /k_O$ (1.6 ±0.3) of linoleic and oleic acids in this study are close to $k_L /k_O$ measured for $O_3$ in laboratory studies, indicating that $O_3$ was the main nighttime oxidants for uFAs in Changzhou City."

---

## Author Comment (AC2)

**Response to comments by Reviewer#2**

We thank the reviewer for the detailed and constructive review comments. Below is our point-by-point response to each comment, marked in blue. Changes made to the main text are marked in red.

**General comments:**

This study aims to characterize the cooking organic aerosol markers evolution in the atmosphere. The study employs online thermal desorption aerosol gas chromatography mass spectrometry (TAG) to analyse the organic component of $PM_{2.5}$ from urban Changzhou, China. The study identified and attributed saturated fatty acids, unsaturated fatty acids, and oxidative decomposition products of unsaturated fatty acids as the molecular marker of cooking sources. Additionally, the decomposition of these markers was estimated using an established ageing parameterisation model, and the contribution of the cooking source to $PM_{2.5}$ was estimated using a positive matrix factorisation model.

The online molecular analysis employed by this study gives a great deal of data that, unfortunately not being optimally used in the analysis and discussion. The study adopted and modified analysis methods by Wang et al. (2020) and Wang and Yu (2021). However, the modification is not entirely justified, raising more questions about the results and discussion. Finally, the association of fatty acids as cooking source molecular markers need more evidence/analysis that considers the other sources, such as biomass burning and marine aerosol.

The manuscript is well written, with some typos that can be improved after thorough checking. The topic presented in this study fits within the scope of the Atmospheric Chemistry and Physics journal. My main concerns are the insufficient evidence to support the cooking organic aerosol molecular marker characterisation and their atmospheric decomposition. In summary, I recommend the Editor reconsider the manuscript after major revisions

**Response:** We thank the reviewer for the comment. We have conducted further data analysis and carefully revised the manuscript to make sure that: (1) the association of fatty acids as cooking source molecular makers is well clarified with more evidence and analysis; (2) the cooking organic aerosol molecular maker characterization and their atmospheric decomposition is well supported by further data analysis; (3) typos are corrected by a thorough language check. Revisions are highlighted in yellow in the revised paper. It should also be noted that in the revised manuscript,

palmitic acid is replaced with stearic acid as a reference compound. In brief, the denominator in the calculation of the ratio of unsaturated fatty acids/saturated fatty acids (uFAs/sFAs) as well as oxidative decomposition products /saturated fatty acids (ODPs/sFAs) is stearic acid. In this manuscript, we reviewed the molecular marker fingerprint of fatty acids from different sources (palmitic acid to stearic acid ratio, P/S) and the relationship between P/S and the ratio of oleic acid /stearic acid (O/S) in organic molecular markers emitted from different cooking types (Fig.4). We also explored the profiles of P/S and O/S in FAs from sea spray aerosol emissions and concluded that sea spray aerosol was not a major contributor to FAs in urban areas during the observation period. We do not include the impact of aging sea spray aerosol on non-coastal cities, where the impact is negligible.

**Specific comments:**

1. Although following previous studies, The Methods section needs explanation and clarifications, which are as listed below.

a). The PM$_{2.5}$ sampling method (Lines 94-96: flow rate, residence time, type of collection matrix, etc)

b). The post-sampling analysis (Lines 100-102: brief procedures). For example, an hourly sampling will result in 24 samples at maximum. An hour of sampling plus 1.5-hour post processing before starting a new sampling sequence will result in less than 12 samples per day.

**Response:** We have revised the Methods section. Please refer to Lines 93-110.

**Lines 93-110:** "Quantification of hourly speciated organic markers was achieved using TAG. The operation details and data quality have been described in our previous work (Wang et al., 2020; Zhang et al., 2021). The sampling and analysis sequence of the TAG system includes four steps: (a) PM$_{2.5}$ sampling and synchronous gas chromatography-mass spectrometry (GC-MS) analysis of the previous sample; (b) loading of the internal standards (IS) from the standards (STD) reservoir to a thermal desorption cell; (c) derivatization and thermal desorption of analytes on the collection and thermal desorption (CTD) cell and subsequent preconcentration of the analytes in focusing trap (FT); and (d) loading of analytes into the GC column for GC-MS analysis. The following is a detailed description. Ambient air was sampled at a flow rate of 8.5-9.5 L/min through a cyclone with PM$_{2.5}$ cutting size (BGI Inc., Waltham, MA), a Nafion dryer (PERMA PURE, MD-700-24S-3) to remove

moisture, and then through a carbon denuder (model: ADI-DEN2) to remove volatile organics. The sampled particles were collected on the CTD cell at 30°C for 60 min, followed by derivatization and thermal desorption for 8 min as the temperature of the CTD cell increases to 300°C in 2 min and maintains for 6 min, during which a 10 mL/min helium purge flow combined with a 40 mL/min derivatization flow with N-methyl- N-(trimethylsilyl) trifluoroacetamide (MSTFA) flow through for 8 min. Subsequently, the FT was heated to 300°C in 2 min and kept at 300°C for 10 min, transferring the analytes onto the GC column head (DB-5MS, size 30 m × 0.25 μm × 0.25 μm) by carrier gas. After GC separation, the target organics were sent to the MS detector for quantification. The GC-MS analysis duration for each sample was 60 min while collection of the next sample the CTD cell starts. With the current TAG instrumental set-up, samples were collected every even hour. The post-sampling steps, including in-situ derivatization, thermal desorption, GC-MS analysis, and standby step, took 2 h, thus producing 12 samples per day."

2. The identification of cooking aerosol markers needs improvement. Attributing all Fatty Acids as cooking aerosol markers is misleading because they are also emitted by other sources, i.e., biomass burning and traffic. The study used a similar measurement method (TAG) to Wang et al. (2020) and is referred to it in the Methods section. That means this study measured other organic molecules, such as sugars (levoglucosan) and PAHs, similar to Wang et al. (2020) study. Moreover, the Supplement Section S2 Figure S7a shows source profiles composed of small and long-chain acids, PAHs, and sugars, suggesting that these data are available for discussion. These other organic groups would improve the identification of cooking source contribution to the total fatty acids emissions.

**Response:** We thank the reviewer for the comment. We agree that fatty acids are emitted by various sources including cooking aerosol, biomass burning and traffic, etc. We have inserted related discussions into the manuscript. We agree that other organic molecules including long-chain acids, sugars, PAHs, etc are also obtained during the same field campaign. We have incorporated these data analysis to improve the identification of cooking source contribution.

(a). Lines 71-74: They are not only markers for cooking/culinary emissions. They can also be emitted by biomass burning, vehicles, and plants (Rogge et al., 1991, 1993, 1998; Simoneit, 2002). Since there are a couple of potential emission sources, discuss how previous studies differentiate

them (Ho et al., 2015, Wang et al., 2020). Ref: Simoneit (2002, https://doi.org/10.1016/S0883-2927(01)00061-0), Rogge et al. (1998, https://doi.org/10.1021/es960930b), Rogge et al. (1991, https://doi.org/10.1021/es00018a015), Rogge et al. (1993, https://doi.org/10.1021/es00041a007), Ho et al. (2015, doi:10.5194/acp-15-3111-2015)

**Response:** Thanks for the comment. We have incorporated related discussions about the various potential sources (For example, biomass burning, vehicle exhaust, coal combustion and sea spray aerosol, etc.) of fatty acids in the urban environment to the manuscript, please refer to Lines 218-231:

**Lines 218-231:** "The main sources of fatty acid-like substances in the atmospheric environment of the study area can be discerned on the basis of characteristic ratios between fatty acids emitted from different sources (Fig.4) (He et al., 2004; Pei et al., 2016; Rogge et al., 1993; Zhao et al., 2015; Zhao et al., 2007). The palmitic acid to stearic acid (P/S) ratios observed in this study had a range between 0.49 and 3.08 (average value: 1.49), significantly lower than those associated with residential coal combustion and industrial coal combustion, while partially overlapping those from biomass burning, vehicle exhaust and sea spray aerosol (Bikkin et al., 2019; Cai et al., 2017; Ho et al., 2015; Zhang et al., 2008; Zhang et al., 2007). Ho et al. (2015) studied urban areas in Beijing where fatty acid concentrations were elevated during traffic restrictions compared to non-restricted periods, suggesting that motor vehicle exhaust is not a significant source of fatty acids in urban areas. In the study of Simoneit (2002), no oleic acid was detected in organic molecular substances from biomass burning. The oleic acid/stearic acid (O/S) ratio from sea spray aerosol samples is 0.16 (Bikkin et al., 2019), which is obviously lower than the ambient data in this study (1.4). Thus, it is reasonable to conclude that biomass burning, vehicle exhaust and sea spray were insignificant sources of fatty acids in urban Changzhou during the observation in this study. Especially during the dinner period, when the O/S ratio was significantly higher and close to the ratio in the organics emitted from traditional culinary types in the Yangtze River Delta region."

(b). What are EP1, EP2, etc., in Figure 2. A summary table of EP1, EP2, etc., parameters can be added to the Supplement. Additionally, plotting fatty acids in log scale make the time trends incomparable to the other parameters plotted in linear scale (Figure 2). What are the reasons behind separating EP2 and EP3 (Figures 2 and 4)? Instead of grouping and averaging the measurement into

day measurement (D1, D2, D3, etc.), time series of TFAs/OC for EP1, EP2, etc. would show the actual trend and give better insight into the transformation of the fatty acids (Lines 265-269).

**Response:** Thanks for the comments. After careful consideration and re-analysis, we agree with the reviewer that limited scientific conclusions can be drawn from the manuscript Fiugre7. We have therefore reorganized the manuscript. Specifically, the second half of section 3.3 (original lines 265-288) and Figure 7 of the original manuscript were deleted, and the previous section (original lines 248-263 and Table 2) was retained and moved to **subsection 3.1 (New manuscript, lines 183-201 and table 2)**. In addition, we removed the information of Episode in Figure 2 and changed the logarithmic Y axis to an arithmetic form.

(c). Section 3.1: How TFAs and FAs are differentiated here? TFAs (Line 168) are similar to FAs (Line 176). Be consistent with using fatty acids or FAs, and FAs or TFAs. Additionally, Considering the other sources of fatty acids, the missing of lunchtime raises a question (Lines 180-185). Could the fatty acids come from biomass burning (residential heating) at night and vehicle emissions in the morning? Wang et al. (2020) observed a small increase in the daytime. Additionally, back trajectory analysis suggests CL#2 and CL#4 airmasses came from the sea (Line 223, Figure 5), suggesting a marine aerosol potential contribution to fatty acids.

**Response:** "TFAs" is the sum of the concentrations of five fatty acids (myristic acid, palmitic acid, stearic acid, oleic acid and linoleic acid). We have replaced all of the "FAs" with "TFAs" to make it consistent throughout the manuscript. We reviewed the molecular marker fingerprinting of fatty acids from different sources (palmitic acid to stearic acid ratio, P/S) and the relationship between P/S and the ratio of oleic acid /stearic acid (O/S) in organic molecular markers emitted from different cooking types (Fig.4). Oleic acid at similar concentrations to palmitic and stearic acid was observed in our study. However in the study of Simoneit (2002), no oleic acid was detected in organic molecular substances from biomass burning, and Ho et al. (2015) studied urban areas in Beijing where fatty acid concentrations were elevated during traffic restrictions compared to non-restricted periods, suggesting that motor vehicle exhaust is not a significant source of fatty acids in urban areas. In addition, Fig.3 and Fig.11 also show that peak contributions of fatty acids and cooking sources are also observed at lunchtime. Therefore, we believe that biomass burning and vehicle exhaust were not the main sources of fatty acids in urban areas. We also explored the profiles of P/S

and O/S in fatty acids from sea spray aerosol emissions, and concluded that sea spray aerosol is not a major contributor to fatty acids in urban areas. Please refer to Lines 209-213 and 216-231.

**Lines 209-213:** "Figure 3(b) shows the contribution of various fatty acids to OC. When the influence of the boundary layer height change was eliminated, the proportion of the five fatty acids and TFAs in OC at noon had a weaker peak, which was still smaller than that during the morning and evening mealtimes. In conclusion, the apparent peaks of TFAs at the dinner time provide strong evidence for source contribution to air pollution from local cooking emissions."

**Lines 216-231:** "Fatty acids in urban atmospheres are influenced by various anthropogenic (e.g., biomass burning, vehicle exhaust) (Hays et al., 2002; Schauer et al., 2001; Simoneit, 2002; Wang et al., 2009) and biogenic sources (Oliveira et al., 2007; Rogge et al., 2006). The main sources of fatty acid-like substances in the atmospheric environment of the study area can be discerned on the basis of characteristic ratios between fatty acids emitted from different sources (Fig.4) (He et al., 2004; Pei et al., 2016; Rogge et al., 1993; Zhao et al., 2015; Zhao et al., 2007). The palmitic acid to stearic acid (P/S) ratios observed in this study had a range between 0.49 and 3.08 (average value: 1.49), significantly lower than those associated with residential coal combustion and industrial coal combustion, while partially overlapping those from biomass burning, vehicle exhaust and sea spray aerosol (Bikkin et al., 2019; Cai et al., 2017; Ho et al., 2015; Zhang et al., 2008; Zhang et al., 2007). Ho et al. (2015) studied urban areas in Beijing where fatty acid concentrations were elevated during traffic restrictions compared to non-restricted periods, suggesting that motor vehicle exhaust is not a significant source of fatty acids in urban areas. In the study of Simoneit (2002), no oleic acid was detected in organic molecular substances from biomass burning. The oleic acid/stearic acid (O/S) ratio from sea spray aerosol samples is 0.16 (Bikkin et al., 2019), which is obviously lower than the ambient data in this study (1.4). Thus, it is reasonable to conclude that biomass burning, vehicle exhaust and sea spray were insignificant sources of fatty acids in urban Changzhou during the observation in this study. Especially during the dinner period, when the O/S ratio was significantly higher and close to the ratio in the organics emitted from traditional culinary types in the Yangtze River Delta region."

(d). Based on Figures 4 and 3, the cooking source seems to contribute at night only. In the morning, the fatty acids could be contributed from non-cooking or long-range transport sources, of which the

latter can be a combination of sources. Additionally, Oleic/Palmitic in Figure 4 is peaking at noontime, which could be associated with fresh cooking emissions at lunchtime. Instead of individual species analysis, cluster analysis of the ratios of the fatty acids could give further insight into their atmospheric evolution.

**Response:** We have modified the figure to include diurnal trends of FAs/OC (Fig.3). The corresponding descriptions in the manuscript have also been modified. We agree with the reviewer and cited several papers in the manuscript to illustrate the sources of fatty acids in $PM_{2.5}$ in urban areas. Ultimately, it was obtained that evening and nighttime fatty acids are mainly from cooking sources. For daytime hours, we were not able to quantify the contribution of various emission sources and long-range transport to fatty acids. The oxidative degradation of oleic acid emitted from fresh cooking sources in the midday hours has also been explained in the manuscript. Please refer to Lines 205-211 and 299-305:

**Lines 205-211:** "Fatty acids showed a clear diurnal variation, with two peaks observed at around 6:00 and 20:00 local time, respectively, and the dinner time peak was especially prominent. In contrast to the previous observations in Shanghai, no peak was observed at lunchtime. The relatively higher boundary layer during the daytime, facilitated the diffusion of pollutants. The weaker oxidation of uFAs emitted at night made the fatty acid concentration peaks more pronounced at dinner time (Wang et al., 2020). Figure 3(b) shows the contribution of various fatty acids to OC. When the influence of the boundary layer height change was eliminated, the proportion of the five fatty acids and TFAs in OC at noon had a weaker peak, which was still smaller than that during the morning and evening mealtimes."

**Lines 299-305:** "Fig.8(a) shows the diurnal variation of ozone, oleic acid, and ODPs. The ozone concentration started to rise in the morning (06:00) and peaked in the late afternoon (14:00). The diurnal trend of oleic acid was opposite to that of ozone. The diurnal trend of ODPs was also different from oleic acid, the small peak of ODPs was found at around 12:00 in the daytime, which was earlier than that of ozone. At the same time, oxidative decomposition caused significant decrease in the concentration of oleic acid until the dinner time when large amounts of fresh emissions enter the atmosphere again. The decreasing rate of oleic acid concentration slowed down around noon, probably because of fresh cooking emission at lunch time."

[Figure]

**Fig.3**. Diurnal variation of five FAs and TFAs during the observation period.

3. This study used parameterisation by Wang and Yu (2021) to estimate the fatty acids atmospheric ageing. Wang and Yu (2021) developed the relative rate constant for the ambient fatty acids based on C18:0 stearic acid as the reference molecule for normalisation.

(a). Any reason for choosing Palmitic Acid instead of Stearic Acid (Lines 132-133)? Wang 2020 plotted Oleic/Stearic or Linoleic/Stearic vs Palmitic/Stearic to investigate the ageing of cooking markers. Is there a particular reason for plotting Oleic/Palmitic or Linoleic/Palmitic versus Palmitic/Stearic (Lines 194-202, Figure 4)? Using different denominators would result in different ageing interpretations. However, it could be good to test the effect of Stearic Acid and Palmitic Acid as the ageing reference to the analysis.

**Response:** Thanks for the comments. In order to make the results more scientific and reasonable, modifications were made in the revised manuscript to keep consistent with Wang et al. (2020), and other studies. In the revised manuscript, palmitic acid was replaced with stearic acid as a reference. In brief, the denominator in the calculation of unsaturated fatty acids/saturated fatty acids (uFAs/sFAs) as well as oxidative decomposition products /saturated fatty acids (ODPs/sFAs) is stearic acid.

(b). Lines 292-293, Figure 8: The plot does not show negative linear correlation as inferred in the text and figure. The curves could be potentially negative exponential for Azelaic or Oxononanoic/Palmitic vs Oleic/Palmitic, but not linear. Moreover, there is no R or $R^2$ value for the regression plots. The R or $R^2$ value is important to assess the association between the two ratios/parameters. Another method is calculating the p-value to show the statistical significance of the two ratios.

(c). What is the reason for plotting Figure 9 in a 2-power scale instead of a linear scale? If these

axes are correlation values for X9/P vs O/P, Figure 9 should plot R2values of each ratio.

**Response to (b) and (c):** We adjusted the section "Atmospheric aging of cooking markers" by merging the original Figure 8 and Figure 9. In Fig.8 (the revised manuscript), the logarithmic form has been used for the y-axis to better distinguish the differences in ODPs/Stearic acid under different gas clusters. No correlation analysis was done in Fig.8 (in the revised manuscript) such that a t-test was not performed. However, significance tests and the p-value estimation was performed in other sections (Fig.9 and Fig.10 in the new manuscript) of this manuscript to make the results more convincing. Please refer to Lines 316-322:

**Lines 316-322:** "Fig.8(b) to (d) show the relationship between ODPs / stearic acid ratio and oleic acid/stearic acid. In CL#2 and CL#4, 9-oxononanoic acid / stearic acid ratio is larger than that in CL#1 and CL#3, and azelaic acid /stearic acid ratio have the same characteristic. The nonanoic acid / stearic acid ratio is not well characterized, probably because most of the nonanoic acid is present in the gas phase. Bikkina et al. (2019) found that the O/S ratio exhibited a nonlinear (power) inverse relationship with azelaic acid in remote marine aerosols. This feature was not found in this study, which is possibly due to the single source class of fatty acids and ODPs in remote marine areas, the diversity of emission sources in urban areas, and their vulnerability to transport."

[Figure]

**Fig.8.** Diurnal variation of C9 products and oleic acid in environmental samples compared to $O_3$ (a);

Correlation of C9 products azelaic acid (b), 9-oxononanoic acid (c), and nonanoic acid (d) with oleic acid.

4. This study used PMF analysis to estimate cooking source contribution to $PM_{2.5}$. However, this study has not adequately discussed the PMF analysis leading to the conclusion of cooking source contribution. It is mentioned that this study would only briefly identify each source factor. However, a discussion of the PMF analysis is still needed to support the conclusion on cooking aerosol contribution to $PM_{2.5}$. The PMF analysis can go to the Supplement as done in this manuscript, but it needs more information. The authors should provide, at a minimum, the correlation coefficient of factor solution profiles and time series and the observed and predicted cooking aerosol markers. Further information, such as the Base Model Displacement Error Method and Bootstrapping analysis, is also important to explore the rotational ambiguity and assess the uncertainty that arises from random errors in the dataset, respectively (e.g., Almeida et al. (2020, https://doi.org/10.1016/j.envpol.2020.115199). Lastly, considering the cooking contribution is smaller than biomass burning, and vehicle exhaust (Figure 12), fatty acids discussed here might not be mainly emitted by cooking activities.

**Response:** Thanks for the constructive comments. Detailed analysis of PMF results have been added to the manuscript and supporting information, including the identification of each source factor, the contribution of the source to $PM_{2.5}$, and the analysis and description of the time variation (including diurnal variation) of each source factor. In addition, table S3 shows the summary of error estimation diagnostics from bootstrap (BS), displacement (DISP), and bootstrap combined with displacement (BS–DISP) for the PMF. And, in the "Response No.2 (a) and (c)" we concluded that biomass burning was not the main source of fatty acids, and in the PMF source allocation results, biomass burning factor was mainly tracers of Levoglucosan and mannosan, and only trace fatty acids were allocated to the modified source. Cooking descriptions is also included in the main manuscript. Please refer to Lines 372-378, 380-391 and Section S2 (table S2 and Fig. S5):

**Lines 372-378:** "Source apportionment of $PM_{2.5}$ in this field campaign yielded 10 sources, including three secondary sources (secondary sulfate, secondary nitrate and SOA, respectively) and seven primary emission sources (cooking, biomass burning, coal combustion, vehicle exhaust, industrial emissions, dust and fire working, respectively). A detailed description of the identification of each

PMF-resolved source factor is shown in section S2. Briefly, secondary source factors account for the largest share of $PM_{2.5}$ (the total was 56.9%, of which secondary nitrate contributes up to 34.4%), and primary emissions contributed to 43.1% of total $PM_{2.5}$ (Fig.11). Among the primary source factors, industry makes the largest contribution to PM2.5 mass concentration (9.9%)."

**Lines 380-391:** "······the cooking factor was dominated by sFAs and uFAs during the monitoring period, accounting for 4.6% of the total $PM_{2.5}$. The concentration of cooking source and its contribution to total $PM_{2.5}$ also showed a clear diurnal variation, with two peaks at around 6:00 and 20:00, respectively, especially at the dinner time. The contribution of cooking to $PM_{2.5}$ concentration during mealtime increased significantly compared with other periods, reaching 7.8% at dinner time. The mean concentration of cooking aerosol in the polluted period was estimated to be 4.0 μg/m$^3$, which was 5.3 times higher than that in the clean period (0.75 μg/m$^3$). The variation was similar to that of fatty acids. The factor profiles of the 10-factor constrained run of PMF are shown in section S2 and Figure S5, together with the time series of contributions from individual source factors. Overall, we estimated that cooking accounted for 5.8% of the total $PM_{2.5}$ during the pollution period, which was 1.9 times greater than that of 3.0% during the clean period. During the whole observation period, the cooking factor contributes only a small part of $PM_{2.5}$, but it accounts for 8.1% of the total OC, indicating the importance of cooking emissions to organic matter, which is a significant source of organic pollution in urban areas."

**Lines 60-84 in the Supporting Information:** "Table S2 lists the input ($PM_{2.5}$ and its components) in the PMF modeling. The preferential input species for PMF are those with high abundance and source specific (Norris et al., 2014). Generally, organic markers with lower volatility and lower reactivity were selected. Figure S5 shows the individual source profiles of the 10 factors resolved in the PMF (a) and time series of individual factor contributions (b). Figure 6 shows the diurnal variation in individual source factors resolved by MM-PMF. Table S3 shows the summary of error estimation diagnostics from bootstrap (BS), displacement (DISP), and bootstrap combined with displacement (BS–DISP) for the PMF base run. Generally, BS and DISP results indicated robust PMF solutions. However, BS–DISP results showed higher uncertainties which may be due to the limited sample size in the study. It should be noted that secondary nitrate and secondary sulfate showed the lowest BS mappings and a high chance of mixing with the vehicle exhaust, industry emission and coal combustion factors. Here, we only briefly present the identification of each source

factor.

A total of 10 factors are identified. Among them, seven are primary sources, they are industrial emission, biomass burning, vehicle exhaust, coal combustion, dust, cooking and fireworking. Three secondary sources, namely, secondary nitrate, secondary sulfate and SOA factor (Li et al., 2020; Wang et al., 2017).

Secondary nitrate factor is identified by high contributions of nitrate and ammonium. The secondary sulfate factor is characterized by high loadings of sulfate and ammonium. The SOA factor is characterized by high loadings of an anthropogenic SOA tracer (phthalic acid), isoprene SOA tracer (2-methylglyceric acid) and α-pinene SOA tracers (3-hydroxyglutaric acid, pinic acid and cis-pinonic acid) (Wang et al., 2017). The profile of industrial emission contains high loadings of Cr, Zn, Fe and Mn (Men et al., 2019; Pant and Harrison, 2013). Industry activities related to steel production, plating, and metallurgy often emit a large amount of these metallic elements. Biomass burning is identified by high loadings of levoglucosan and mannosan (Feng et al., 2013; Wang et al., 2019). The seventh factor contains a high abundance of n-alkanes and hopanes, and is identified to be vehicle exhaust (Pant and Harrison, 2013; Wang et al., 2017). Coal combustion is identified by high loadings of Se, As and Pb (Chen et al., 2013; Wang et al., 2017), and the dust factor is distinguished by crustal elements (ions) Ca, Si, and Ti. The cooking factor is distinguished by fatty acids (oleic acid, palmitic acid and stearic acid) (Li et al., 2020). The fireworking factor is identified by high loadings of flammable metals such as Mg, Cu and Ba, etc."

[Figure]

**Fig.11**. Comparison of individual factor contributions to $PM_{2.5}$ (a) and OC (b); diurnal variation of

cooking source (c).

[Figure]

**Fig. S5.** Individual source profiles of the 10 factors resolved in the PMF (a) and time series of individual factor contributions (b).

**Technical comments**

a). Lines 227-228: The component of ODPs haven't been explained in the text.

**Response:** Thanks for pointing this out. ODPs has been explained in the revised manuscript.

**Lines 272-274**: "The total concentrations of sFAs, uFAs and uFAs' oxidative decomposition products (ODPs, in this study, ODPs includes azelaic acid, nonanoic acid and 9-oxonononanoic acid)⋯⋯⋯⋯"

b). Figure 5: Add a legend to explain the coloured lines for the Cluster.

**Response:** Thanks for the suggestion, we have revised the figure.

[Figure]

**Fig.8.** Sources for each air mass during the sampling period. The colored lines in the map show the contribution of each directional air mass source to the total trajectory as resolved by the TrajStat model.

c). Lines 260-262: Add reference studies or ratios for TFAs/OC from other sources.

**Response:** Thanks. This section is supplemented in the supporting information as Table S1, which includes the source profiles of organic molecular markers emitted by cooking source, biomass burning, vehicle exhaust and other source factors.

Table S1 Contribution of TFAs to total OC from different sources (ng/μg).

| Sources | Min | Max | Avg. | Std | References |
|---------|-----|-----|------|-----|------------|
| Biomass Burning | 6.6 | 25.8 | 14.5 | 5.8 | (Hays et al., 2002; Schauer et al., 2001; Zhang et al., 2007) |
| Coal Combustion | 3.9 | 29.9 | 11.6 | 9.9 | (Zhang et al., 2008) |
| Vehicle Exhaust | 0.32 | 6.2 | 2.6 | 2.1 | (Cai et al., 2017) |
| Cooking | 27.7 | 82.2 | 55.9 | 21.4 | (Pei et al., 2016; Zhao et al., 2015; Zhao et al., 2007) |

d). Figure 8a: What does the y-axis Contribution mean?

**Response:** Fig.8(a) y-axis shows the normalized parameters, that is, the concentration of each substance at different times divided by its mean concentration, which is used as the standard to show the diurnal variation rule.

e). Line 311: Correlation as in correlation coefficient values or relationship between Y and X axes?

**Response:** Thanks for the comment. In the original figure, the correlation value and the fitted formula represent the relationship between the parameters represented by the X and Y axes.

f). Lines 391-302, Figure S7: There is no time series or diurnal plot of the PMF factor solution in the Supplement.

**Response:** Thanks for the suggestion. Detailed analysis of PMF results has been inserted to the manuscript and supporting information, including the identification of each source factor, the contribution of the source to $PM_{2.5}$, and the analysis and description of the time variation (including diurnal variation) of each source factor. (See "**Response to Comment 4".**)

---

## Author Response (AR2)

**Response to reviewers' comments**

**Referee #1:** "The authors have responded well and in detail to each of the comments provided by the reviewers, which has resulted in substantial improvements to the text and figures. I believe this manuscript will now provide a valuable addition to the literature. I would therefore recommend publication in ACP, after the minor comments outlined below have been addressed."

We thank all the reviewers for the constructive comments which have helped to improve the paper substantially. We have carefully addressed the minor comments outlined during the second-round review. Below is our point-by-point response to each comment, marked in blue. Changes made to the main text are marked in red. Revisions made to the manuscript are highlighted.

The main concern raised by both reviewers was the limited evidence provided to support the authors' assertion that the predominant source of fatty acids in this environment is cooking. The revised manuscript contains more detailed consideration of alternatives (eg biomass burning, traffic emissions, coal burning, sea spray). Nevertheless, I still feel there could be more acknowledgement that there could be at least some influence from other sources. For example, in lines 227-228, the authors dismiss the possibility of a sea spray contribution based on the oleic acid/stearic acid ratio from a single study (Bikkina et al., 2019). While this certainly provides evidence in favour of the authors' hypothesis, I would argue that it is not strong enough to conclusively rule out the potential contribution, as the authors have done here.

**Response**: We agree with the reviewer that there are diverse sources of fatty acids besides cooking, such as biomass burning, traffic emissions, coal burning, sea spray etc. Based on the source profiles from literature, and the observed data from TAG, as well as the anthropogenic source characteristics in Changzhou, we conclude that the predominant source of fatty acids in the Changzhou City is likely to be from cooking sources. In lines 227-228, we mentioned that "The oleic acid/stearic acid (O/S) ratio from sea spray aerosol samples is 0.16 (Bikkin et al., 2019), which is obviously lower than the ambient data in this study (1.4)." From this data as well as location of Changzhou (located in the inland YRD, far away from the sea), we believe that sea spray contribution to fatty acid in Changzhou is negligible. We agree that it is not appropriate to conclusively rule out the potential contribution from other sources, therefore, in the revised manuscript, we have softened the language through out the mansucript.

The new Fig. 4 is a valuable addition as it provides more context. It could be interesting to include data points from non-cooking sources in Fig. 4b, if they are available, as well as the campaign average. This could provide supporting evidence for the claim that the majority of the observed fatty

acids originate from cooking.

Response: We thank the reviewer for the constructive comment. We have added data of non-cooking sources in Fig. 4b, and inserted the references to the figure title of Fig.4.

[Figure]

Figure 4. Ratio of fatty acids (P/S) in organic molecular substances emitted directly from different sources (a); Ratio of fatty acids (P/S vs O/S) emitted by different types of cooking sources and non-cooking sources (b). (Bikkin et al., 2019; Cai et al., 2017; Hays et al., 2002; He et al., 2004; Oliveira et al., 2007; Pei et al., 2016; Rogge et al., 1993; Schauer et al., 2001, 2002; Simoneit, 2002; Zhang et al., 2008; Zhao et al., 2007).

**Referee #3:**

In this study, the authors used TAG techniques to study fatty acid concentrations in the atmosphere, including saturated fatty acids, unsaturated fatty acids, and their oxidative decomposition products. The authors attribute these fatty acid species to emissions from cooking. Their higher time resolution chemical speciation with TAG represents a detailed, valuable dataset, and I appreciate their use of backward trajectories and PMF to understand chemical sources. However, I echo the previous review comments that some of the conclusions here still aren't fully supported by the data. I can see that the authors did carefully address reviewer comments from the first round. I would suggest that the authors further soften the language in the text throughout the manuscript to account for the fact that there is significant uncertainty in the sources of the primary and secondary products observed here. This is a result of field measurements being very challenging to interpret with mixed sources contributing to any one measurement, and I think this needs to be better acknowledged and reflected in the manuscript. I think the manuscript is acceptable for publication in Atmospheric Chemistry and Physics following a careful review of how each conclusion is stated.

Response: We thank the reviewer for the detailed and constructive comments and suggestions. We have carefully checked the manuscript to make sure that the conclusions are clear and could be well supported by the data presented. We agree that it is not appropriate to conclusively rule out the potential contribution from other sources, therefore, in the revised manuscript, we have softened the language. Below is our point-by-point response to each comment, marked in blue. Changes made to the main text are marked in red. Revisions made to the manuscript are highlighted.

**Introduction:**

1. Line 51: I would be careful with this phrasing, so that you do not attribute all lung cancer to exposure to cooking fumes.

Response: Thanks for the comment. We have revised the description. Please refer to Lines 52-53.

**Lines 52-53**: "The carcinogenic risk analysis suggested that the potentially adverse health effects induced by cooking sources should not be ignored."

2. Line 62: write out "hydroxyl" and "" and then use the acronyms after you've defined them in words.

Response: We have defined these words in the new manuscript.

**Line 61:** hydroxyl (OH)

**Line 71**: "OH and nitrate ($NO_3$) radicals".

3. I would consider re-organizing the introduction so that you talk about TAG methods (and why

online measurements are important) after introducing cooking emissions more – so perhaps could move lines 55-69 to after lines 70-77.

Response: Thanks for the comment. We have adjusted the order of the TAG methods and cooking emissions in the introduction section. Please refer to Lines 54-78.

**Lines 54-78**: "Cooking is an important source contributor to $PM_{2.5}$, especially in urban environments. Cooking sources have recently received increasing attention, but they are largely an uncontrolled source of $PM_{2.5}$. Saturated fatty acids (sFAs) and unsaturated fatty acids (uFAs)····················································The aging of POA markers under atmospheric conditions, however, is still far from being properly understood with few field observations performed in this topic compared to laboratory studies (Bertrand et al., 2018a; Bertrand et al., 2018b). The high timely-resolved observations would help to fill this gap."

**Methodology:**

4.  Line 85: please be more specific about what you mean by "its main chemical constituents as well as organic markers"

Response: We added specific information about "its main chemical constituents as well as organic markers". Please refer to Lines 86-88.

**Lines 86-88**: "Gaseous pollutants, $PM_{2.5}$ and its main chemical constituents (water soluble ions, carbon components and elements, etc.) as well as organic markers (alkanes, hopanes, polycyclic aromatic hydrocarbons, sugars, alcohols and organic acids, etc.) ············."

5.  Can you comment on any fatty acid losses to your instrument, i.e., in the cyclone, dryer, or denuder?

Response: Thanks for the constructive comment. Regarding the possible material loss during the observation of $PM_{2.5}$ by the TAG instrument, Zhao (2012) conducted relevant experiments at the early stage of the instrument development and explored the collection efficiency of the TAG instrument for different carbon numbers of organic matter and the amount of loss of particles of different particle sizes at the denuder respectively. The results demonstrate that the accuracy of the data obtained from TAG's observations of atmospheric pollutants is high and can accurately reflect the pollution levels of atmospheric pollutants.

**Reference:** Zhao, Y.: Organic Aerosol Sources and Chemistry: Insights from Development and Application of In-Situ Thermal Desorption Gas Chromatograph for Semi-Volatile Organic Compounds (SV-TAG), 2012.

**Results and discussion:**

6. Line 164-165: It would help to list out these instruments specifically in your methods section. I would also suggest moving the paragraph where you describe each of these instruments in the SI to your main text. It is a short paragraph and it would help readers understand the measurements you took.

Response: Thanks for the suggestion. We have moved the paragraph regarding description of instruments to the main text, please refer to Lines 89-96.

**Lines 89-96**: "The meteorological parameters were obtained from a meteorological monitor (WXT520, VAISALA Inc., FL). O3 and NO2 were measured by ozone analyzer (49i-PS, Thermo Fisher Scientific, US) and NOx analyzer (MODEL450i, Thermo Fisher Scientific, US), respectively. $PM_{2.5}$ mass concentration was measured by an online particulate matter monitor (BAM1020. Met One Inc., US); the concentration of the carbon components (Organic carbon, OC; Elemental carbon, EC) was measured using semi-continuous OC/EC analyzer (RT-4, Sunset Laboratory Inc, US) (Nicolosi et al., 2018; Zhang et al., 2017b); water soluble ions were measured by MARGA ionic online analyzer (ADI2080, Metrohm, CHN) (Makkonen et al., 2012) and elements were measured by an atmospheric elements online monitor (EHM-X200, Tianrui, CHN) (Makkonen et al., 2012)."

7. Axes on Figure 2 are very hard to read – can you put more space between each panel? Also your axis breaks are very small, can you make them larger? And finally, can you put a number below your axis break (not just above), so it is more clear what the values are right before and after the break?

Response: We have revised Figure 2 accordingly.

[Figure]

8. Line 180: specify what time of day "dinner time" is

Response: We have clarified the "dinner time" in the manuscript, please refer to Lines 187-188.

**Lines 187-188:** "It revealed that the composition of PM$_{2.5}$ could dramatically change, especially during the dinner time (18:00-20:00)."

9. Line 183: What meteorological conditions are present in "clean" and "polluted" periods? Are there more emissions during the "polluted period" or simply different meteorological conditions?

Response: We inserted meteorological conditions in the manuscript. Please refer to Lines 192-195 and Table 2. Generally, the meteorological conditions are unfavorable during the "polluted" periods, showing relatively low wind speed and high humidity.

**Lines 192-196:** "Table 2 shows the mean values of PM$_{2.5}$, OC, TFAs concentrations and meteorological conditions during the clean (PM$_{2.5}$ <35μg/m$^3$) and polluted periods (PM$_{2.5}$ ≥ 35μg/m$^3$). Generally, the meteorological conditions during the polluted period are unfavourable compared to the clean period, showing lower wind speed and higher humidity. The ratios of WS, T and RH during the polluted period to the clean period are 0.9, 1.0 and 1.1 respectively."

**Table 2. PM$_{2.5}$ concentration, organic carbon fraction, fatty acids concentration and meteorological conditions during clean and polluted periods.**

| Species | Clean period | Polluted period | Polluted/clean |
|---|---|---|---|
| PM$_{2.5}$ (μg/m$^3$) | 28.29 ± 5.27 | 62.86 ± 25.67 | 2.2 |
| OC (μg/m$^3$) | 4.05 ± 1.09 | 8.00 ± 5.23 | 2.0 |

| | | | |
|---|---|---|---|
| TFAs (ng/m$^3$) | 35.28 ± 28.17 | 147.06 ± 281.66 | 4.2 |
| sFAs (ng/m$^3$) | 21.60 ± 14.91 | 92.05 ± 162.75 | 4.3 |
| uFAs (ng/m$^3$) | 13.68 ± 14.22 | 55.53 ± 133.82 | 4.1 |
| TFAs/PM$_{2.5}$ (ng/μg) | 1.24 ± 0.91 | 1.95 ± 2.85 | 1.6 |
| TFAs/OC (ng/μg) | 9.52 ± 7.79 | 15.33 ± 14.59 | 1.6 |
| WS (m/s) | 1.23 ± 0.45 | 1.14 ± 0.54 | 0.9 |
| T (℃) | 10.77 ± 4.22 | 10.99 ± 4.68 | 1.0 |
| RH (%) | 53.41 ± 17.49 | 56.33 ± 18.55 | 1.1 |

10. Line 210: How did you eliminate boundary layer height? Is that what is shown in b?

Response: Thanks for the constructive comment. The diurnal pattern shown in Fig.3(a) is very characteristic of a parameter influenced by boundary layer dynamics. However, it is difficult to discern which changes are related to boundary layer changing and which to fresh emissions. Fig.3(b) showed the diurnal patterns of each compound as a fraction of OC, which can eliminate the influence of the boundary layer height in a relative way, so as to give more insight into the chemical changes. To avoid misunderstanding, we have revised the description. Please refer to Lines 221-222.

**Lines 221-222:** From the diurnal patterns, it is shown that the proportion of the five fatty acids and TFAs in OC at noon had a weaker peak, which was still smaller than that during the morning and evening mealtimes.

11. Line 212: I agree, but there are also other active sources during these times, as people heat their homes and drive their vehicles to/from work, so I would be careful with this language here. I don't think it is convincing that these are ONLY cooking emissions. The paragraph from lines 224-231 is valuable (discussing ratios of fatty acids from difference sources). Figure 4 is also valuable, depicting this information. But cooking ratios overlap so much with pretty much all other source profiles, so I think you need to be careful how much you emphasize that these are cooking derived-emissions. Also, what references are contributing to figure 4? Please cite them in the figure itself.

Response: We thank the reviewer for the comments and suggestions. We reorganized and softened the language in Line 212, Please refer to Lines 222-224. The source references of the data in the figure are specifically cited in the manuscript, readers can find the references from Lines 227-244. Since the number of cited references is large, and the names of the references cannot be well shown in the figure one by one, we have inserted the references to the figure title of Fig.4.

**Lines 222-224**: In conclusion, the apparent peaks of TFAs at the dinner time provide evidence for

major source contribution to air pollution from local cooking emissions, although there are mix sources including vehicle exhaust, coal combustion, etc.

**Lines 227-243:** "Fatty acids in urban atmospheres are influenced by various anthropogenic (e.g., biomass burning, vehicle exhaust) (Hays et al., 2002; Schauer et al., 2001; Simoneit, 2002; Wang et al., 2009) and biogenic sources (Oliveira et al., 2007; Rogge et al., 2006). The main sources of fatty acid-like substances in the ambient air of the study area can be discerned on the basis of characteristic ratios between fatty acids emitted from different sources (Fig.4) (He et al., 2004; Pei et al., 2016; Rogge et al., 1993; Zhao et al., 2015; Zhao et al., 2007). The palmitic acid to stearic acid (P/S) ratios observed in this study ranges between 0.49 and 3.08 (average value: 1.49), significantly lower than those associated with residential coal combustion and industrial coal combustion, while partially overlapping those from biomass burning, vehicle exhaust and sea spray aerosol (Bikkin et al., 2019; Cai et al., 2017; Ho et al., 2015; Zhang et al., 2008; Zhang et al., 2007). Ho et al. (2015) investigated urban areas in Beijing where fatty acid concentrations were elevated during traffic restrictions compared to non-restricted periods, suggesting that motor vehicle exhaust is not the largest source of fatty acids in urban areas. The information on fatty acid emissions from biomass burning source is closer to that of cooking sources (Hays et al., 2002; Schauer et al., 2001; Zhang et al., 2008), however, in the study of Simoneit (2002), no oleic acid was detected in organic molecular substances from biomass burning. The oleic acid/stearic acid (O/S) ratio from sea spray aerosol samples is 0.16 (Bikkin et al., 2019), which is obviously lower than the ambient data in this study (1.4). Hence, during the observation in this study, vehicle exhaust and sea spray were not the most important sources of fatty acids emissions in urban Changzhou. Especially during the dinner period, when the O/S ratio was significantly higher and close to the ratio in the organics emitted from traditional culinary types in the Yangtze River Delta region."

**Fig.4**

[Figure]

Figure 4. Ratio of fatty acids (P/S) in organic molecular substances emitted directly from different sources (a); Ratio of fatty acids (P/S vs O/S) emitted by different types of cooking sources and non-cooking sources (b). (Bikkin et al., 2019; Cai et al., 2017; Hays et al., 2002; He et al., 2004; Oliveira et al., 2007; Pei et al., 2016; Rogge et al., 1993; Schauer et al., 2001, 2002; Simoneit, 2002; Zhang et al., 2008; Zhao et al., 2007).

12. Line 246: Can you prove this with ozone reactivities? Linoleic acid has an extra double bond than oleic acid so it makes sense it would degrade in the atmosphere more rapidly.

Response: Thanks for the constructive comment. Moise & Rudich (2002) and Thornberry & Abbatt (2004) illustrate the degradation rates of two unsaturated fatty acids in terms of ozone loss kinetics and the specific reactions of oleic and linoleic acids with ozone. We have added literature citations related to this in the manuscript. Please refer to Lines 260-263 and References (Moise and Rudich, 2002; Thornberry and Abbatt, 2004).

Lines 260-262: "The results of Moise and Rudich (2002) showed that the reactant activity is directly related to the concentration of unsaturated bonds, with linoleic acid having an extra double bond than oleic acid, indicating that linoleic acid is more easily degraded than oleic acid (Moise and Rudich, 2002; Thornberry and Abbatt, 2004).

References: Moise, T., and Rudich, Y.: Reactive uptake of ozone by aerosol-associated unsaturated fatty acids: Kinetics, mechanism, and products, Journal of Physical Chemistry A, 106(27), 6469–6476, doi.org/10.1021/jp025597e, 2002.

Thornberry, T., and Abbatt, J. P. D.: Heterogeneous reaction of ozone with liquid unsaturated fatty acids: detailed kinetics and gas-phase product studies, PCCP, 6(1), 84-93, doi:10.1039/b310149e,

2004.

13. Figure 5 caption: Please provide more details on how this cooking source profile was determined.

Response: Thanks for the constructive comment. We have made corresponding citations to the cooking source literature in the manuscript, along with additional descriptions in the caption of Figure 5. Please refer to Lines 254-265 and Fig.5.

**Lines 254-265:** "Fig.5 shows the O/S ratios and linoleic acid/ stearic acid (L/S) versus P/S, respectively. The average value of P/S was 1.49±0.49, which was within the range of cooking source profile values measured from direct emissions from different restaurants and cooking types (1.3-8.1) (He et al., 2004; Pei et al., 2016; Schauer et al., 2002; Zhao et al., 2007), and similar to the ratio of P/S in atmospheric $PM_{2.5}$ in Shanghai (1.9) (Li et al., 2020; Wang et al., 2020). In this study, the O/S ratio (1.4 ± 1.1) of the ambient samples was overall in the range of the cooking source profile (3.6 ± 1.6), while the L/S ratio of 0.25 ± 0.31 was slightly lower than the cooking source profile values (2.9 ± 1.8) (He et al., 2004; Pei et al., 2016; Schauer et al., 2002; Zhao et al., 2007). The results of Moise and Rudich (2002) showed that the reactant activity is directly related to the concentration of unsaturated bonds, with linoleic acid has an extra double bond than oleic acid, indicating that linoleic acid is more easily degraded than oleic acid (Moise and Rudich, 2002; Thornberry and Abbatt, 2004). The O/S ratio of the ambient samples in this study was higher than those measured in Beijing (0.65) (He et al., 2004) from January to October and in Shanghai (0.83) (Li et al., 2020; Wang et al., 2020) during winter."

**Figure 5.** The oleic/ stearic acid and linoleic/ stearic acid ratios compared to the palmitic/stearic acid ratio (a); diurnal variation in the ratio of oleic (linoleic) acid to stearic acid concentration (b). (The cooking source profile values were measured from direct emissions from different restaurants and cooking types.) (He et al., 2004; Pei et al., 2016; Schauer et al., 2002; Zhao et al., 2007).

14. Line 272: should say "Figure 6"

Response: Thanks for pointing this out. "Fig.5" has been revised to "Fig.6". Please refer to Line 291.

**Line 291:** "The concentrations of sFAs, uFAs and their oxidation products under each cluster are shown in Fig.6."

15. Lines 286-290: What about primary sources of these ODPs from the ocean? I don't think these are solely attributable to oxidation, and I suggest softening the language here.

Response: Thanks for the constructive comment. We have revised the language description and

added references. Please refer to Lines 307-309.

**Lines 307-309:** "··········for example, marine heterotrophic bacteria releases sFAs and uFAs into the water column, however the mono/polyunsaturated fatty acids (e.g., oleic acid, linoleic acid) in seawater rapidly oxidizes to form initially oxocarboxylic acids, azelaic acid etc. (Bikkin et al., 2019); ··········"

16. Figure 8: The color of $O_3$ in the legend and in the figure is different

Response: Thanks for pointing this out. We have made the appropriate changes. Please refer to Figure 8.

[Figure]

17. Line 302: How can you be certain that this is just driven by oxidative decomposition, not dilution or lower emissions? The language here needs to be softened.

Response: Thanks for the constructive comment. We have reorganized the language and made additions to the manuscript. Please refer to Lines 323-325.

**Lines 323-325:** "····· ····· At the same time, oxidative decomposition, atmospheric dilution and lower emissions caused significant decrease in the concentration of oleic acid until night when large amounts of fresh emissions enter the atmosphere again. ····· ······"

18. Line 305: How can you be certain that the ONLY sources of these two species is oleic acid ozonolysis? The language here needs to be softened to account for the other possible sources.

Response: Thanks for the comment. We have revised accordingly, please refer to Lines 326-331.

**Lines 326-331:** "C9 ω-oxo acid and diacids (e.g., nonanoic acid, 9-oxononanoic acid and azelaic acid) in the atmospheric environment originate from plant volatilization, combustion emissions, and

cooking processes (Kawamura et al., 2013; Tian et al., 2020), and they were established in chamber studies as major atmospheric oxidation products from uFAs ozonolysis (Kawamura et al., 2013; Moise and Rudich, 2002; Thornberry and Abbatt, 2004). The diurnal variations of Nonanoic acid and 9-oxononanoic acid were similar and both peaked around noon, while the production of 9-oxononanoic acid and azelaic acid are in competition (Thornberry and Abbatt, 2004)."

19. Line 334: What about the impact of plume dilution during this time? In addition to oxidative decay?

**Response**: Thanks for the constructive comment. We studied the oxidative decay of unsaturated fatty acids using the method of Donahue et al (2005) and Wang & Yu (2021). And among the quantified sFA and uFA cooking markers, palmitic acid was selected as the reference molecule for normalization. Wang and Yu (2021) suggested that using the concentration ratio (uFAs / sFAs) eliminates the interference from atmospheric dilution and deposition. Please refer to Lines 146-150 and References: Donahue et al (2005) and Wang & Yu (2021).

**Lines 146-150:** "$C_i$ and $C_s$ are the particle-phase concentration of species i and sFAs, respectively. Among the quantified sFA and uFA cooking markers, palmitic acid was selected as the reference molecule for normalization. Using the concentration ratio eliminates the interference from atmospheric dilution and deposition. Fitting the ambient $C_i/C_s$ data versus t with an exponential function provides an estimate for k, the effective pseudo-first order decay rate ($h^{-1}$). $k_{ri}$ is the second-order reaction rate constant of species i against an oxidant. $C_{OX}$ is the average oxidant concentration in the aerosol."

**References:** Donahue, N. M., Robinson, A. L., Huff Hartz, K. E., Sage, A. M., and Weitkamp, E. A.: Competitive oxidation in atmospheric aerosols: The case for relative kinetics, Geophys. Res. Lett., 32(16), doi:10.1029/2005gl022893, 2005.

Wang, Q. Q., and Yu, J. Z.: Ambient Measurements of Heterogeneous Ozone Oxidation Rates of Oleic, Elaidic, and Linoleic Acid Using a Relative Rate Constant Approach in an Urban Environment, Geophys. Res. Lett., 48(19), doi:10.1029/2021GL095130, 2021.